# WEAKLY SUPERVISED DISENTANGLEMENT WITH GUARANTEES

**Rui Shu**[†][*]**, Yining Chen**[†]**, Abhishek Kumar**[‡]**, Stefano Ermon**[†]**& Ben Poole**[‡]
[†]Stanford University, [‡]Google Brain
[†]{ruishu,cynnjjs,ermon}@stanford.edu
[‡]{abhishk,pooleb}@google.com

## ABSTRACT

Learning *disentangled* representations that correspond to factors of variation in real-world data is critical to interpretable and human-controllable machine learning. Recently, concerns about the viability of learning disentangled representations in a purely unsupervised manner has spurred a shift toward the incorporation of weak supervision. However, there is currently no formalism that identifies when and how weak supervision will guarantee disentanglement. To address this issue, we provide a theoretical framework to assist in analyzing the disentanglement guarantees (or lack thereof) conferred by weak supervision when coupled with learning algorithms based on distribution matching. We empirically verify the guarantees and limitations of several weak supervision methods (restricted labeling, match-pairing, and rank-pairing), demonstrating the predictive power and usefulness of our theoretical framework.

## 1 INTRODUCTION

Many real-world datasets can be intuitively described via a data-generating process that first samples an underlying set of interpretable factors, and then—conditional on those factors—generates an observed data point. For example, in image generation, one might first sample the object identity and pose, and then render an image with the object in the correct pose. The goal of disentangled representation learning is to learn a representation where each dimension of the representation corresponds to a distinct *factor of variation* in the dataset (Bengio et al., 2013). Learning such representations that align with the underlying factors of variation may be critical to the development of machine learning models that are explainable or human-controllable (Gilpin et al., 2018; Lee et al., 2019; Klys et al., 2018).

In recent years, disentanglement research has focused on the learning of such representations in an *unsupervised* fashion, using only independent samples from the data distribution without access to the true factors of variation (Higgins et al., 2017; Chen et al., 2018a; Kim & Mnih, 2018; Esmaeili et al., 2018). However, Locatello et al. (2019) demonstrated that many existing methods for the unsupervised learning of disentangled representations are brittle, requiring careful supervision-based hyperparameter tuning. To build robust disentangled representation learning methods that do not require large amounts of supervised data, recent work has turned to forms of weak supervision (Chen & Batmanghelich, 2019; Gabbay & Hoshen, 2019). Weak supervision can allow one to build models that have interpretable representations even when human labeling is challenging (e.g., hair style in face generation, or style in music generation). While existing methods based on weakly-supervised learning demonstrate empirical gains, there is no existing formalism for describing the theoretical guarantees conferred by different forms of weak supervision (Kulkarni et al., 2015; Reed et al., 2015; Bouchacourt et al., 2018).

In this paper, we present a comprehensive theoretical framework for weakly supervised disentanglement, and evaluate our framework on several datasets. Our contributions are several-fold.

1. We formalize weakly-supervised learning as distribution matching in an extended space.

---

[*]Work done during an internship at Google Brain.

2. We propose a set of definitions for disentanglement that can handle correlated factors and are inspired by many existing definitions in the literature (Higgins et al., 2018; Suter et al., 2018; Ridgeway & Mozer, 2018).

3. Using these definitions, we provide a conceptually useful and theoretically rigorous calculus of disentanglement.

4. We apply our theoretical framework of disentanglement to analyze three notable classes of weak supervision methods (restricted labeling, match pairing, and rank pairing). We show that although certain weak supervision methods (e.g., style-labeling in style-content disentanglement) do not guarantee disentanglement, our calculus can determine whether disentanglement is guaranteed when multiple sources of weak supervision are combined.

5. Finally, we perform extensive experiments to systematically and empirically verify our predicted guarantees.[1]

## 2 FROM UNSUPERVISED TO WEAKLY SUPERVISED DISTRIBUTION MATCHING

Our goal in disentangled representation learning is to identify a latent-variable generative model whose latent variables correspond to ground truth factors of variation in the data. To identify the role that weak supervision plays in providing guarantees on disentanglement, we first formalize the model families we are considering, the forms of weak supervision, and finally the metrics we will use to evaluate and prove components of disentanglement.

We consider data-generating processes where $S \in \mathbb{R}^n$ are the factors of variation, with distribution $p^*(s)$, and $X \in \mathbb{R}^m$ is the observed data point which is a deterministic function of $S$, i.e., $X = g^*(S)$. Many existing algorithms in *unsupervised* learning of disentangled representations aim to learn a latent-variable model with prior $p(z)$ and generator $g$, where $g(Z) \stackrel{d}{=} g^*(S)$. However, simply matching the marginal distribution over data is not enough: the learned latent variables $Z$ and the true generating factors $S$ could still be entangled with each other (Locatello et al., 2019).

To address the failures of unsupervised learning of disentangled representations, we leverage weak supervision, where information about the data-generating process is conveyed through additional observations. By performing distribution matching on an augmented space (instead of just on the observation $X$), we can provide guarantees on learned representations.

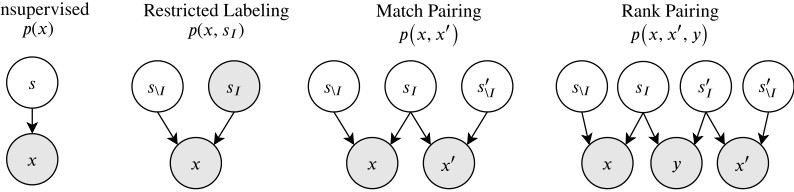

Figure 1: Augmented data distributions derived from weak supervision. Shaded nodes denote observed quantities, and unshaded nodes represent unobserved (latent) variables.

We consider three practical forms of weak supervision: restricted labeling, match pairing, and rank pairing. All of these forms of supervision can be thought of as augmented forms of the original joint distribution, where we partition the latent variables in two $S = (S_I, S_{\setminus I})$, and either observe a subset of the latent variables or share latents between multiple samples. A visualization of these augmented distributions is presented in Figure 1, and below we detail each form of weak supervision.

In **restricted labeling**, we observe a subset of the ground truth factors, $S_I$ in addition to $X$. This allows us to perform distribution matching on $p^*(s_I, x)$, the joint distribution over data and observed factors, instead of just the data, $p^*(x)$, as in unsupervised learning. This form of supervision is often leveraged in style-content disentanglement, where labels are available for content but not style (Kingma et al., 2014; Narayanaswamy et al., 2017; Chen et al., 2018b; Gabbay & Hoshen, 2019).

---

[1]Code available at https://github.com/google-research/google-research/tree/master/weak_disentangle

**Match Pairing** uses paired data, $(x, x')$ that share values for a known subset of factors, $I$. For many data modalities, factors of variation may be difficult to explicitly label. Instead, it may be easier to collect pairs of samples that share the same underlying factor (e.g., collecting pairs of images of different people wearing the same glasses is easier than defining labels for style of glasses). Match pairing is a weaker form of supervision than restricted labeling, as the learning algorithm no longer depends on the underlying value $s_I$, and only on the indices of shared factors $I$. Several variants of match pairing have appeared in the literature (Kulkarni et al., 2015; Bouchacourt et al., 2018; Ridgeway & Mozer, 2018), but typically focus on groups of observations in contrast to the paired setting we consider in this paper.

**Rank Pairing** is another form of paired data generation where the pairs $(x, x')$ are generated in an i.i.d. fashion, and an additional indicator variable $y$ is observed that determines whether the corresponding latent $s_i$ is greater than $s_i'$: $y = \mathbf{1}\{s_i \geq s_i'\}$. Such a form of supervision is effective when it is easier to compare two samples with respect to an underlying factor than to directly collect labels (e.g., comparing two object sizes versus providing a ruler measurement of an object). Although supervision via ranking features prominently in the metric learning literature (McFee & Lanckriet, 2010; Wang et al., 2014), our focus in this paper will be on rank pairing in the context of disentanglement guarantees.

For each form of weak supervision, we can train generative models with the same structure as in Figure 1, using data sampled from the ground truth model and a distribution matching objective. For example, for match pairing, we train a generative model $(p(z), g)$ such that the paired random variable $(g(Z_I, Z_{\setminus I}), g(Z_I, Z'_{\setminus I}))$ from the generator matches the distribution of the corresponding paired random variable $(g^*(S_I, S_{\setminus I}), g^*(S_I, S'_{\setminus I}))$ from the augmented data distribution.

## 3 Defining Disentanglement

To identify the role that weak supervision plays in providing guarantees on disentanglement, we introduce a set of definitions that are consistent with our intuitions about what constitutes "disentanglement" and amenable to theoretical analysis. Our new definitions decompose disentanglement into two distinct concepts: consistency and restrictiveness. Different forms of weak supervision can enable consistency or restrictiveness on subsets of factors, and in Section 4 we build up a calculus of disentanglement from these primitives. We discuss the relationship to prior definitions of disentanglement in Appendix A.

### 3.1 Decomposing Disentanglement into Consistency and Restrictiveness

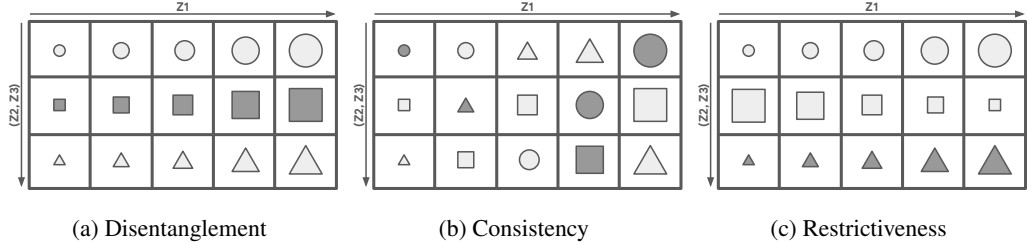

(a) Disentanglement       (b) Consistency       (c) Restrictiveness

Figure 2: Illustration of disentanglement, consistency, and restrictiveness of $z_1$ with respect to the factor of variation *size*. Each image of a shape represents the decoding $g(z_{1:3})$ by the generative model. Each column denotes a fixed choice of $z_1$. Each row denotes a fixed choice of $(z_2, z_3)$. A demonstration of consistency versus restrictiveness on models from `disentanglement_lib` is available in Appendix B.

To ground our discussion of disentanglement, we consider an oracle that generates shapes with factors of variation for *size* ($S_1$), *shape* ($S_2$), and *color* ($S_3$). How can we determine whether $Z_1$ of our generative model "disentangles" the concept of size? Intuitively, one way to check whether $Z_1$ of the generative model disentangles *size* ($S_1$) is to visually inspect what happens as we vary $Z_1$, $Z_2$, and $Z_3$, and see whether the resulting visualizations are consistent with Figure 2a. In doing so, our visual inspection checks for two properties:

1. When $Z_1$ is fixed, the *size* ($S_1$) of the generated object never changes.
2. When only $Z_1$ is changed, the change is restricted to the *size* ($S_1$) of the generated object, meaning that there is no change in $S_j$ for $j \neq 1$.

We argue that disentanglement decomposes into these two properties, which we refer to as *generator consistency* and *generator restrictiveness*. Next, we formalize these two properties.

Let $\mathcal{H}$ be a hypothesis class of generative models from which we assume the true data-generating function is drawn. Each element of the hypothesis class $\mathcal{H}$ is a tuple $(p(s), g, e)$, where $p(s)$ describes the distribution over factors of variation, the generator $g$ is a function that maps from the factor space $\mathcal{S} \in \mathbb{R}^n$ to the observation space $\mathcal{X} \in \mathbb{R}^m$, and the encoder $e$ is a function that maps from $\mathcal{X} \rightarrow \mathcal{S}$. $S$ and $X$ can consist of both discrete and continuous random variables. We impose a few mild assumptions on $\mathcal{H}$ (see Appendix I.1). Notably, we assume every factor of variation is exactly recoverable from the observation $X$, i.e. $e(g(S)) = S$.

Given an oracle model $h^* = (p^*, g^*, e^*) \in \mathcal{H}$, we would like to learn a model $h = (p, g, e) \in \mathcal{H}$ whose latent variables disentangle the latent variables in $h^*$. We refer to the latent-variables in the oracle $h^*$ as $S$ and the alternative model $h$'s latent variables as $Z$. If we further restrict $h$ to only those models where $g(Z) \stackrel{d}{=} g^*(S)$ are equal in distribution, it is natural to align $Z$ and $S$ via $S = e^* \circ g(Z)$. Under this relation between $Z$ and $S$, our goal is to construct definitions that describe whether the latent code $Z_i$ disentangles the corresponding factor $S_i$.

**Generator Consistency.** Let $I$ denote a set of indices and $p_I$ denote the generating process

$$z_I \sim p(z_I) \tag{1}$$

$$z_{\backslash I}, z'_{\backslash I} \stackrel{\text{iid}}{\sim} p(z_{\backslash I} \mid z_I). \tag{2}$$

This generating process samples $Z_I$ once and then conditionally samples $Z_I$ twice in an i.i.d. fashion. We say that $Z_I$ is consistent with $S_I$ if

$$\mathbb{E}_{p_I} \| e_I^* \circ g(z_I, z_{\backslash I}) - e_I^* \circ g(z_I, z'_{\backslash I}) \|^2 = 0, \tag{3}$$

where $e_I^*$ is the oracle encoder restricted to the indices $I$.

Intuitively, Equation (3) states that, for any fixed choice of $Z_I$, resampling of $Z_{\backslash I}$ will not influence the oracle's measurement of the factors $S_I$. In other words, $S_I$ is *invariant* to changes in $Z_{\backslash I}$. An illustration of a generative model where $Z_1$ is consistent with *size* ($S_1$) is provided in Figure 2b. A notable property of our definition is that the prescribed sampling process $p_I$ does not require the underlying factors of variation to be statistically independent. We characterize this property in contrast to previous definitions of disentanglement in Appendix A.

**Generator Restrictiveness.** Let $p_{\backslash I}$ denote the generating process

$$z_{\backslash I} \sim p(z_{\backslash I}) \tag{4}$$

$$z_I, z'_I \stackrel{\text{iid}}{\sim} p(z_I \mid z_{\backslash I}). \tag{5}$$

We say that $Z_I$ is restricted to $S_I$ if

$$\mathbb{E}_{p_{\backslash I}} \| e_{\backslash I}^* \circ g(z_I, z_{\backslash I}) - e_{\backslash I}^* \circ g(z'_I, z_{\backslash I}) \|^2 = 0. \tag{6}$$

Equation (6) states that, for any fixed choice of $Z_{\backslash I}$, resampling of $Z_I$ will not influence the oracle's measurement of the factors $S_{\backslash I}$. In other words, $S_{\backslash I}$ is *invariant* to changes in $Z_I$. Thus, changing $Z_I$ is restricted to modifying only $S_I$. An illustration of a generative model where $Z_1$ is restricted to *size* ($S_1$) is provided in Figure 2c.

**Generator Disentanglement.** We now say that $Z_I$ disentangles $S_I$ if $Z_I$ is consistent with *and* restricted to $S_I$. If we denote consistency and restrictiveness via Boolean functions $C(I)$ and $R(I)$, we can now concisely state that

$$D(I) := C(I) \land R(I), \tag{7}$$

where $D(I)$ denotes whether $Z_I$ disentangles $S_I$. An illustration of a generative model where $Z_1$ disentangles *size* ($S_1$) is provided in Figure 2a. Note that while size increases monotonically with $Z_1$ in the schematic figure, we wish to clarify that monotonicity is unrelated to the concepts of consistency and restrictiveness.

## 3.2 Relation to Bijectivity-Based Definition of Disentanglement

Under our mild assumptions on $\mathcal{H}$, distribution matching on $g(Z) \stackrel{d}{=} g(S)$ combined with generator disentanglement on factor $I$ implies the existence of two invertible functions $f_I$ and $f_{\setminus I}$ such that the alignment via $S = e^* \circ g(Z)$ decomposes into

$$\begin{bmatrix} S_I \\ S_{\setminus I} \end{bmatrix} = \begin{bmatrix} f_I(Z_I) \\ f_{\setminus I}(Z_{\setminus I}) \end{bmatrix}. \tag{8}$$

This expression highlights the connection between disentanglement and *invariance*, whereby $S_I$ is only influenced by $Z_I$, and $S_{\setminus I}$ is only influenced by $Z_{\setminus I}$. However, such a bijectivity-based definition of disentanglement does not naturally expose the underlying primitives of *consistency* and *restrictiveness*, which we shall demonstrate in our theory and experiments to be valuable concepts for describing disentanglement guarantees under weak supervision.

## 3.3 Encoder-Based Definitions for Disentanglement

Our proposed definitions are asymmetric—measuring the behavior of a generative model against an oracle encoder. So far, we have chosen to present the definitions from the perspective of a learned generator $(p, g)$ measured against an oracle encoder $e^*$. In this sense, they are *generator-based* definitions. We can also develop a parallel set of definitions for *encoder-based* consistency, restrictiveness, and disentanglement within our framework simply by using an oracle generator $(p^*, g^*)$ measured against a learned encoder $e$. Below, we present the encoder-based perspective on consistency.

**Encoder Consistency.** Let $p_I^*$ denote the generating process

$$s_I \sim p^*(s_I) \tag{9}$$

$$s_{\setminus I}, s'_{\setminus I} \stackrel{\text{iid}}{\sim} p^*(s_{\setminus I}, \mid s_I). \tag{10}$$

This generating process samples $S_I$ once and then conditionally samples $S_I$ twice in an i.i.d. fashion. We say that $S_I$ is consistent with $Z_I$ if

$$\mathbb{E}_{p_I^*} \| e_I \circ g^*(s_I, s_{\setminus I}) - e_I \circ g^*(s_I, s'_{\setminus I}) \|^2 = 0. \tag{11}$$

We now make two important observations. First, a valuable trait of our encoder-based definitions is that one can check for encoder consistency / restrictiveness / disentanglement *as long as one has access to match pairing data from the oracle generator*. This is in contrast to the existing disentanglement definitions and metrics, which require access to the ground truth factors (Higgins et al., 2017; Kumar et al., 2018; Kim & Mnih, 2018; Chen et al., 2018a; Suter et al., 2018; Ridgeway & Mozer, 2018; Eastwood & Williams, 2018). The ability to check for our definitions in a weakly supervised fashion is the key to why we can develop a theoretical framework using the language of consistency and restrictiveness. Second, encoder-based definitions are tractable to measure when testing on synthetic data, since the synthetic data directly serves the role of the oracle generator. As such, while we develop our theory to guarantee *both* generator-based and the encoder-based disentanglement, all of our measurements in the experiments will be conducted with respect to a learned encoder.

We make three remarks on notations. First, $D(i) := D(\{i\})$. Second, $D(\varnothing)$ evaluates to true. Finally, $D(I)$ is implicitly dependent on either $(p, g, e^*)$ (generator-based) or $(p^*, g^*, e)$ (encoder-based). Where important, we shall make this dependency explicit (e.g., let $D(I ; p, g, e^*)$ denote generator-based disentanglement). We apply these conventions to $C$ and $R$ analogously.

## 4 A Calculus of Disentanglement

There are several interesting relationships between restrictiveness and consistency. First, by definition, $C(I)$ is equivalent to $R(\setminus I)$. Second, we can see from Figures 2b and 2c that $C(I)$ and $R(I)$ do not imply each other. Based on these observations and given that consistency and restrictiveness operate over *subsets* of the random variables, a natural question that arises is whether consistency or restrictiveness over certain sets of variables imply additional properties over other sets of variables.

We develop a calculus for discovering *implied* relationships between learned latent variables $Z$ and ground truth factors of variation $S$ given known relationships as follows.

---

**Calculus of Disentanglement**

**Consistency and Restrictiveness**
$$C(I) \;\not\Longrightarrow\; R(I) \qquad\qquad R(I) \;\not\Longrightarrow\; C(I) \qquad\qquad C(I) \Longleftrightarrow R(\backslash I)$$

**Union Rules**
$$C(I) \wedge C(J) \;\Longrightarrow\; C(I \cup J) \qquad\qquad R(I) \wedge R(J) \;\Longrightarrow\; R(I \cup J)$$

**Intersection Rules**
$$C(I) \wedge C(J) \;\Longrightarrow\; C(I \cap J) \qquad\qquad R(I) \wedge R(J) \;\Longrightarrow\; R(I \cap J)$$

**Full Disentanglement**
$$\bigwedge_{i=1}^{n} C(i) \Longleftrightarrow \bigwedge_{i=1}^{n} D(i) \qquad\qquad \bigwedge_{i=1}^{n} R(i) \Longleftrightarrow \bigwedge_{i=1}^{n} D(i)$$

---

Our calculus provides a theoretically rigorous procedure for reasoning about disentanglement. In particular, it is no longer necessary to prove whether the supervision method of interest satisfies consistency and restrictiveness for each and every factor. Instead, it suffices to show that a supervision method guarantees consistency or restrictiveness for a subset of factors, and then combine multiple supervision methods via the calculus to guarantee full disentanglement. We can additionally use the calculus to uncover consistency or restrictiveness on individual factors when weak supervision is available only for a subset of variables. For example, achieving consistency on $S_{1,2}$ and $S_{2,3}$ implies consistency on the intersection $S_2$. Furthermore, we note that these rules are agnostic to using generator or encoder-based definitions. We defer the complete proofs to Appendix I.2.

## 5   FORMALIZING WEAK SUPERVISION WITH GUARANTEES

In this section, we address the question of whether disentanglement arises from the supervision method or model inductive bias. This challenge was first put forth by Locatello et al. (2019), who noted that unsupervised disentanglement is heavily reliant on model inductive bias. As we transition toward supervised approaches, it is crucial that we formalize what it means for disentanglement to be guaranteed by weak supervision.

**Sufficiency for Disentanglement.** Let $\mathcal{P}$ denote a family of augmented distributions. We say that a weak supervision method $\mathbf{S} : \mathcal{H} \to \mathcal{P}$ is *sufficient* for learning a generator whose latent codes $Z_I$ disentangle the factors $S_I$ if there exists a learning algorithm $\mathcal{A} : \mathcal{P} \to \mathcal{H}$ such that for any choice of $(p^*(s), g^*, e^*) \in \mathcal{H}$, the procedure $\mathcal{A} \circ \mathbf{S}(p^*(s), g^*, e^*)$ returns a model $(p(z), g, e)$ for which both $D(I \,;\, p, g, e^*)$ and $D(I \,;\, p^*, g^*, e)$ hold, and $g(Z) \stackrel{d}{=} g^*(S)$.

The key insight of this definition is that we force the strategy and learning algorithm pair $(\mathbf{S}, \mathcal{A})$ to handle all possible oracles drawn from the hypothesis class $\mathcal{H}$. This prevents the exploitation of model inductive bias, since any bias from the learning algorithm $\mathcal{A}$ toward a reduced hypothesis class $\hat{\mathcal{H}} \subset \mathcal{H}$ will result in failure to handle oracles in the complementary hypothesis class $\mathcal{H} \setminus \hat{\mathcal{H}}$.

The distribution matching requirement $g(Z) \stackrel{d}{=} g^*(S)$ ensures latent code informativeness, i.e., preventing trivial solutions where the latent code is uninformative (see Proposition 6 for formal statement). Intuitively, distribution matching paired with a deterministic generator guarantees invertibility of the learned generator and encoder, enforcing that $Z_I$ cannot encode less information than $S_I$ (e.g., only encoding age group instead of numerical age) and vice versa.

## 6   ANALYSIS OF WEAK SUPERVISION METHODS

We now apply our theoretical framework to three practical weak supervision methods: restricted labeling, match pairing, and rank pairing. Our main theoretical findings are that: (1) these methods can be applied in a targeted manner to provide single factor consistency or restrictiveness guarantees; (2) by enforcing consistency (or restrictiveness) on all factors, we can learn models with strong

disentanglement performance. Correspondingly, Figure 3 and Figure 5 are our main experimental results, demonstrating that these theoretical guarantees have predictive power in practice.

## 6.1 THEORETICAL GUARANTEES FROM WEAK SUPERVISION

We prove that if a training algorithm successfully matches the generated distribution to data distribution generated via restricted labeling, match pairing, or rank pairing of factors $S_I$, then $Z_I$ is guaranteed to be *consistent* with $S_I$:

**Theorem 1.** *Given any oracle $(p^*(s), g^*, e^*) \in \mathcal{H}$, consider the distribution-matching algorithm $\mathcal{A}$ that selects a model $(p(z), g, e) \in \mathcal{H}$ such that:*

1. $(g^*(S), S_I) \overset{d}{=} (g(Z), Z_I)$ (***Restricted Labeling***); or

2. $\left(g^*(S_I, S_{\setminus I}), g^*(S_I, S'_{\setminus I})\right) \overset{d}{=} \left(g(Z_I, Z_{\setminus I}), g(Z_I, Z'_{\setminus I})\right)$ (***Match Pairing***); or

3. $(g^*(S), g^*(S'), \mathbf{1}\{S_I \leq S'_I\}) \overset{d}{=} (g(Z), g(Z'), \mathbf{1}\{Z_I \leq Z'_I\})$ (***Rank Pairing***).

*Then $(p, g)$ satisfies $C(I\,;\,p, g, e^*)$ and $e$ satisfies $C(I\,;\,p^*, g^*, e)$.*

Theorem 1 states that distribution-matching under restricted labeling, match pairing, or rank pairing of $S_I$ guarantees both generator *and* encoder consistency for the learned generator and encoder respectively. We note that while the complement rule $C(I) \implies R(\setminus I)$ further guarantees that $Z_{\setminus I}$ is restricted to $S_{\setminus I}$, we can prove that the same supervision does *not* guarantee that $Z_I$ is restricted to $S_I$ (Theorem 2). However, if we additionally have restricted labeling for $S_{\setminus I}$, or match pairing for $S_{\setminus I}$, then we can see from the calculus that we will have guaranteed $R(I) \wedge C(I)$, thus implying disentanglement of factor $I$. We also note that while restricted labeling and match pairing can be applied on a set of factors at once (i.e. $|I| \geq 1$), rank pairing is restricted to one-dimensional factors for which an ordering exists. In the experiments below, we empirically verify the theoretical guarantees provided in Theorem 1.

## 6.2 EXPERIMENTS

We conducted experiments on five prominent datasets in the disentanglement literature: *Shapes3D* (Kim & Mnih, 2018), *dSprites* (Higgins et al., 2017), *Scream-dSprites* (Locatello et al., 2019), *SmallNORB* (LeCun et al., 2004), and *Cars3D* (Reed et al., 2015). Since some of the underlying factors are treated as nuisance variables in SmallNORB and Scream-dSprites, we show in Appendix C that our theoretical framework can be easily adapted accordingly to handle such situations. We use generative adversarial networks (GANs, Goodfellow et al. (2014)) for learning $(p, g)$ but any distribution matching algorithm (e.g., maximum likelihood training in tractable models, or VI in latent-variable models) could be applied. Our results are collected over a broad range of hyperparameter configurations (see Appendix H for details).

Since existing quantitative metrics of disentanglement all measure the performance of an encoder with respect to the true data generator, we trained an encoder *post-hoc* to approximately invert the learned generator, and measured all quantitative metrics (e.g., mutual information gap) on the encoder. Our theory assumes that the learned generator must be invertible. While this is not true for conventional GANs, our empirical results show that this is not an issue in practice (see Appendix G).

We present three sets of experimental results: (1) Single-factor experiments, where we show that our theory can be applied in a *targeted* fashion to guarantee consistency or restrictiveness of a single factor. (2) Consistency versus restrictiveness experiments, where we show the extent to which single-factor consistency and restrictiveness are correlated even when the models are only trained to maximize one or the other. (3) Full disentanglement experiments, where we apply our theory to fully disentangle all factors. A more extensive set of experiments can be found in the Appendix.

### 6.2.1 SINGLE-FACTOR CONSISTENCY AND RESTRICTIVENESS

We empirically verify that single-factor consistency or restrictiveness can be achieved with the supervision methods of interest. Note there are two special cases of match pairing: one where $S_i$ is

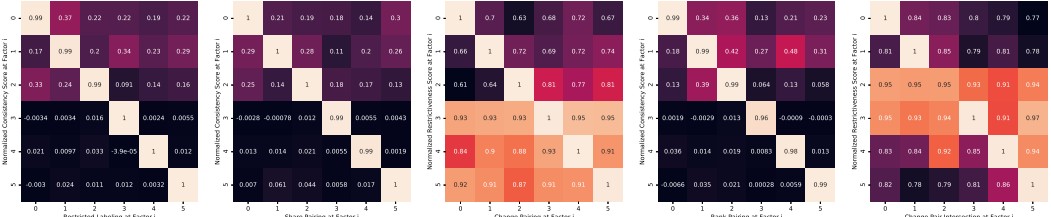

Figure 3: Heatmap visualization of ablation studies that measure either single-factor consistency or single-factor restrictiveness as a function of various supervision methods, conducted on Shapes3D. Our theory predicts the diagonal components to achieve the highest scores. Note that share pairing, change pairing, and change pair intersection are special cases of match pairing.

the only factor that is shared between $x$ and $x'$ and one where $S_i$ is the only factor that is changed. We distinguish these two conditions as *share pairing* and *change pairing*, respectively. Theorem 1 shows that restricted labeling, share pairing, and rank pairing of the $i^{\text{th}}$ factor are each sufficient supervision strategies for guaranteeing consistency on $S_i$. Change pairing at $S_i$ is equivalent to share pairing at $S_{\setminus i}$; the complement rule $C(I) \iff R(\setminus I)$ allows us to conclude that change pairing guarantees restrictiveness. The first four heatmaps in Figure 3 show the results for restricted labeling, share pairing, change pairing, and rank pairing. The numbers shown in the heatmap are the *normalized consistency and restrictiveness scores*. We define the normalized consistency score as

$$\tilde{c}(I\,;p^*,g^*,e) = 1 - \frac{\mathbb{E}_{p_I^*}\|e_I \circ g^*(s_I, s_{\setminus I}) - e_I \circ g^*(s_I, s'_{\setminus I})\|^2}{\mathbb{E}_{s,s' \overset{\text{iid}}{\sim} p^*}\|e_I \circ g^*(s) - e_I \circ g^*(s')\|^2}. \tag{12}$$

This score is bounded on the interval $[0, 1]$ (a consequence of Lemma 1) and is maximal when $C(I\,;p^*,g^*,e)$ is satisfied. This normalization procedure is similar in spirit to the Interventional Robustness score in Suter et al. (2018). The normalized restrictiveness score $\tilde{r}$ can be analogously defined. In practice, we estimate this score via Monte Carlo estimation.

The final heatmap in Figure 3 demonstrates the calculus of intersection. In practice, it may be easier to acquire paired data where multiple factors change simultaneously. If we have access to two kinds of datasets, one where $S_I$ are changed and one where $S_J$ are changed, our calculus predicts that training on both datasets will guarantee restrictiveness on $S_{I \cap J}$. The final heatmap shows six such intersection settings and measures the normalized restrictiveness score; in all but one setting, the results are consistent with our theory. We show in Figure 7 that this inconsistency is attributable to the failure of the GAN to distribution-match due to sensitivity to a specific hyperparameter.

### 6.2.2 CONSISTENCY VERSUS RESTRICTIVENESS

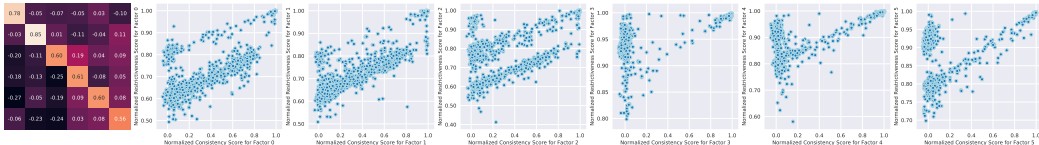

Figure 4: Correlation plot and scatterplots demonstrating the empirical relationship between $\tilde{c}(i)$ and $\tilde{r}(i)$ across all $864$ models trained on Shapes3D.

We now determine the extent to which consistency and restrictiveness are correlated in practice. In Figure 4, we collected all $864$ Shapes3D models that we trained in Section 6.2.1 and measured the consistency and restrictiveness of each model on each factor, providing both the correlation plot and scatterplots of $\tilde{c}(i)$ versus $\tilde{r}(i)$. Since the models trained in Section 6.2.1 only ever targeted the consistency *or* restrictiveness of a single factor, and since our calculus demonstrates that consistency and restrictiveness do not imply each other, one might *a priori* expect to find no correlation in Figure 4. Our results show that the correlation is actually quite strong. Since this correlation is not guaranteed by our choice of weak supervision, it is necessarily a consequence of model inductive

bias. We believe this correlation between consistency and restrictiveness to have been a general source of confusion in the disentanglement literature, causing many to either observe or believe that restricted labeling or share pairing on $S_i$ (which only guarantees consistency) is sufficient for disentangling $S_i$ (Kingma et al., 2014; Chen & Batmanghelich, 2019; Gabbay & Hoshen, 2019; Narayanaswamy et al., 2017). It remains an open question why consistency and restrictiveness are so strongly correlated when training existing models on real-world data.

### 6.2.3 FULL DISENTANGLEMENT

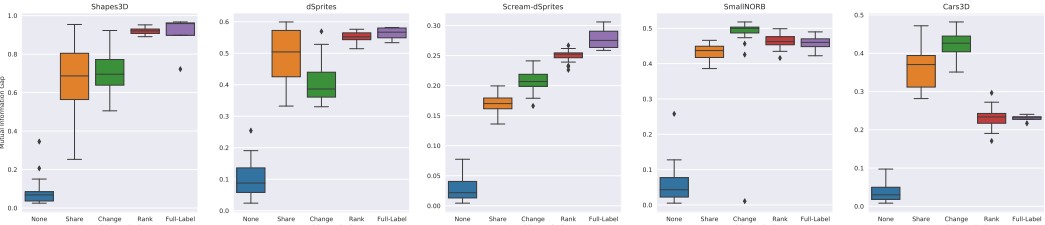

Figure 5: Disentanglement performance of a vanilla GAN, share pairing GAN, change pairing GAN, rank pairing GAN, and fully-labeled GAN, as measured by the mutual information gap across several datasets. A comprehensive set of performance evaluations on existing disentanglement metrics is available in Figure 13.

If we have access to share / change / rank-pairing data for each factor, our calculus states that it is possible to guarantee full disentanglement. We trained our generative model on either complete share pairing, complete change pairing, or complete rank pairing, and measured disentanglement performance via the discretized mutual information gap (Chen et al., 2018a; Locatello et al., 2019). As negative and positive controls, we also show the performance of an unsupervised GAN and a fully-supervised GAN where the latents are fixed to the ground truth factors of variation. Our results in Figure 5 empirically verify that combining single-factor weak supervision datasets leads to consistently high disentanglement scores.

## 7 CONCLUSION

In this work, we construct a theoretical framework to rigorously analyze the disentanglement guarantees of weak supervision algorithms. Our paper clarifies several important concepts, such as consistency and restrictiveness, that have been hitherto confused or overlooked in the existing literature, and provides a formalism that precisely distinguishes when disentanglement arises from supervision versus model inductive bias. Through our theory and a comprehensive set of experiments, we demonstrated the conditions under which various supervision strategies *guarantee* disentanglement. Our work establishes several promising directions for future research. First, we hope that our formalism and experiments inspire greater theoretical and scientific scrutiny of the inductive biases present in existing models. Second, we encourage the search for other learning algorithms (besides distribution-matching) that may have theoretical guarantees when paired with the right form of supervision. Finally, we hope that our framework enables the theoretical analysis of other promising weak supervision methods.

### ACKNOWLEDGMENTS

We would like to thank James Brofos and Honglin Yuan for their insightful discussions on the theoretical analysis in this paper, and Aditya Grover and Hung H. Bui for their helpful feedback during the course of this project.

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

APPENDIX

Our appendix consists of nine sections. We provide a brief summary of each section below.

Appendix A: We elaborate on the connections between existing definitions of disentanglement and our definitions of consistency / restrictiveness / disentanglement. In particular, we highlight three notable properties of our definitions not present in many existing definitions.

Appendix B: We evaluate our consistency and restrictiveness metrics on the 10800 models in the disentanglement_lib, and identify models where consistency and restrictiveness are not correlated.

Appendix C: We adapt our definitions to be able to handle nuisance variables. We do so through a simple modification of the definition of restrictiveness.

Appendix D: We show several additional single-factor experiments. We first address one of the results in the main text that is not consistent with our theory, and explain why it can be attributed to hyperparameter sensitivity. We next unwrap the heatmaps into more informative boxplots.

Appendix E: We provide an additional suite of consistency versus restrictiveness experiments by comparing the effects of training with share pairing (which guarantees consistency), change pairing (which guarantees restrictiveness), and *both*.

Appendix F: We provide full disentanglement results on all five datasets as measured according to six different metrics of disentanglement found in the literature.

Appendix G: We show visualizations of a weakly supervised generative model trained to achieve full disentanglement.

Appendix H: We describe the set of hyperparameter configurations used in all our experiments.

Appendix I: We provide the complete set of assumptions and proofs for our theoretical framework.

## A   CONNECTIONS TO EXISTING DEFINITIONS

Numerous definitions of disentanglement are present in the literature (Higgins et al., 2017; 2018; Kim & Mnih, 2018; Suter et al., 2018; Ridgeway & Mozer, 2018; Eastwood & Williams, 2018; Chen et al., 2018a). We mostly defer to the terminology suggested by Ridgeway & Mozer (2018), which decomposes disentanglement into *modularity*, *compactness*, and *explicitness*. Modularity means a latent code $Z_i$ is predictive of at most one factor of variation $S_j$. Compactness means a factor of variation $S_i$ is predicted by at most one latent code $Z_j$. And explicitness means a factor of variation $S_j$ is predicted by the latent codes via a simple transformation (e.g. linear). Similar to Eastwood & Williams (2018); Higgins et al. (2018), we suggest a further decomposition of Ridgeway & Mozer (2018)'s explicitness into *latent code informativeness* and *latent code simplicity*. In this paper, we omit latent code simplicity from consideration. Since informativeness of the latent code is already enforced by our requirement that $g(Z)$ is equal in distribution to $g^*(S)$ (see Proposition 6), we focus on comparing our proposed concepts of consistency and restrictiveness to modularity and compactness. We make note of three important distinctions.

**Restrictiveness is not synonymous with either modularity or compactness**. In Figure 2c, it is evident the factor of variation *size* is not predictable any individual $Z_i$ (conversely, $Z_1$ is not predictable from any individual factor $S_i$). As such, $Z_1$ is neither a modular nor compact representation of *size*, despite being restricted to *size*. To our knowledge, no existing quantitative definition of disentanglement (or its decomposition) specifically measures restrictiveness.

**Consistency and restrictiveness are invariant to statistically dependent factors of variation**. Many existing definitions of disentanglement are instantiated by measuring the mutual information between $Z$ and $S$. For example, Ridgeway & Mozer (2018) defines that a latent code $Z_i$ to be "ideally modular" if it has high mutual information with a single factor $S_j$ and zero mutual information with all other factors $S_{\setminus j}$. This presents a issue when the true factors of variation themselves are statistically dependent; even if $Z_1 = S_1$, the latent code $Z_1$ would violate modularity if $S_1$ itself has positive mutual information with $S_2$. Consistency and restrictiveness circumvent this issue by relying on conditional resampling. Consistency, for example, only measures the extent to which $S_I$

is *invariant* to resampling of $Z_{\setminus I}$ when conditioned on $Z_I$ and is thus achieved as long as $s_I$ is a function of only $z_I$—irrespective of whether $s_I$ and $s_{\setminus I}$ are statistically dependent. In this regard, our definitions draw inspiration from Suter et al. (2018)'s intervention-based definition but replaces the need for counterfactual reasoning with the simpler conditional sampling. Because we do not assume the factors of variation are statistically independent, our theoretical analysis is also distinct from the closely-related match pairing analysis in Gresele et al. (2019).

**Consistency and restrictiveness arise in weak supervision guarantees**. One of our goals is to propose definitions that are amenable to theoretical analysis. As we can see in Section 4, consistency and restrictiveness serve as the core primitive concepts that we use to describe disentanglement guarantees conferred by various forms of weak supervision.

## B EVALUATING CONSISTENCY AND RESTRICTIVENESS ON DISENTANGLEMENT-LIB MODELS

To better understand the empirical relationship between consistency and restrictiveness, we calculated the normalized consistency and restrictiveness scores on the suite of 12800 models from `disentanglement_lib` for each ground-truth factor. By using the normalized consistency and restrictiveness scores as probes, we were able to identify models that achieve high consistency but low restrictiveness (and vice versa). In Fig. 6, we highlight two models that are either consistent or restrictive for object color on the Shapes3D dataset.

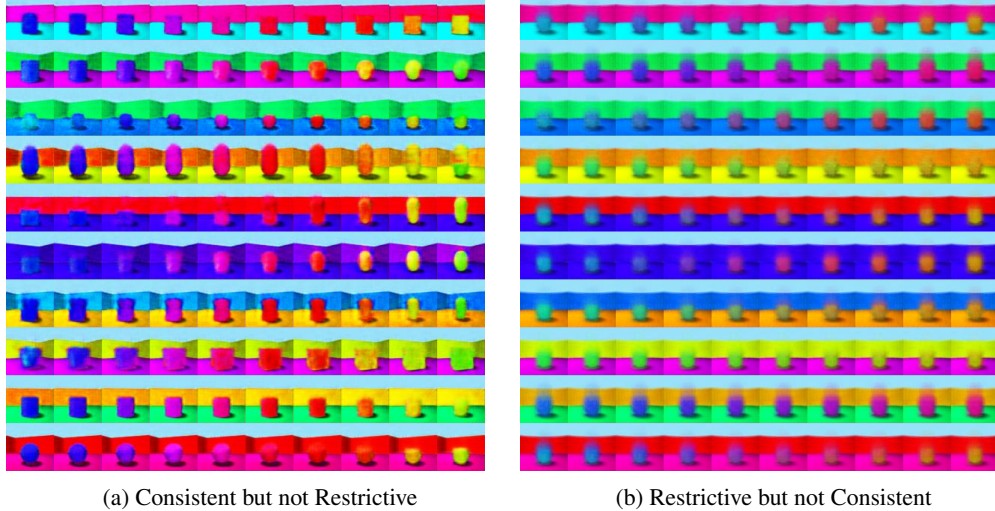

(a) Consistent but not Restrictive        (b) Restrictive but not Consistent

Figure 6: Visualization of two models from `disentanglement_lib` (model ids 11964 and 12307), matching the schematic in Fig. 2. For each panel, we visualize an interpolation along a single latent across rows, with each row corresponding to a fixed set of values for all other factors. In Fig. 6a, we can see that this factor consistenly represents object color, i.e. each column of images has the same object color, but as we move along rows we see that other factors change as well, e.g. object type, thus this factor is not restricted to object color. In Fig. 6b, we see that varying the factor along each row results in changes to object color but to no other attributes. However if we look across columns, we see that the representation of color changes depending on the setting of other factors, thus this factor is not consistent for object color.

## C HANDLING NUISANCE VARIABLES

Our theoretical framework can handle *nuisance variables*, i.e., variables we cannot measure or perform weak supervision on. It may be impossible to label, or provide match-pairing on that factor of variation. For example, while many features of an image are measurable (such as brightness and

coloration), we may not be able to measure certain factors of variation or generate data pairs where these factors are kept constant. In this case, we can let one additional variable $\eta$ act as nuisance variable that captures all additional sources of variation / stochasticity.

Formally, suppose the full set of true factors is $S \cup \{\eta\} \in \mathbb{R}^{n+1}$. We define $\eta$-consistency $C_\eta(I) = C(I)$ and $\eta$-restrictiveness $R_\eta(I) = R(I \cup \{\eta\})$. This captures our intuition that, with nuisance variable, for consistency, we still want changes to $Z_{\setminus I} \cup \{\eta\}$ to not modify $S_I$; for restrictiveness, we want changes to $Z_I \cup \{\eta\}$ to only modify $S_I \cup \{\eta\}$. We define $\eta$-disentanglement as $D_\eta(I) = C_\eta(I) \wedge R_\eta(I)$.

All of our calculus still holds where we substitute $C_\eta(I), R_\eta(I), D_\eta(I)$ for $C(I), R(I), D(I)$; we prove one of the new full disentanglement rule as an illustration:

**Proposition 1.** $\bigwedge_{i=1}^n C_\eta(i) \Longleftrightarrow \bigwedge_{i=1}^n D_\eta(i)$.

*Proof.* On the one hand, $\bigwedge_{i=1}^n C_\eta(i) \Longleftrightarrow \bigwedge_{i=1}^n C(i) \implies C(1:n) \implies R(\eta)$. On the other hand, $\bigwedge_{i=1}^n C(i) \implies \bigwedge_{i=1}^n D(i) \implies \bigwedge_{i=1}^n R(i)$. Therefore $LHS \implies \forall i \in [n], R(i) \wedge R(\eta) \implies R_\eta(i)$. The reverse direction is trivial. $\qquad \square$

In (Locatello et al., 2019), the "instance" factor in SmallNORB and the background image factor in Scream-dSprites are treated as nuisance variables. By Proposition 1, as long as we perform weak supervision on all of the non-nuisance variables (via sharing-pairing, say) to guarantee their consistency with respect to the corresponding true factor of variation, we still have guaranteed full disentanglement despite the existence of nuisance variable and the fact that we cannot measure or perform weak supervision on nuisance variable.

# D SINGLE-FACTOR EXPERIMENTS

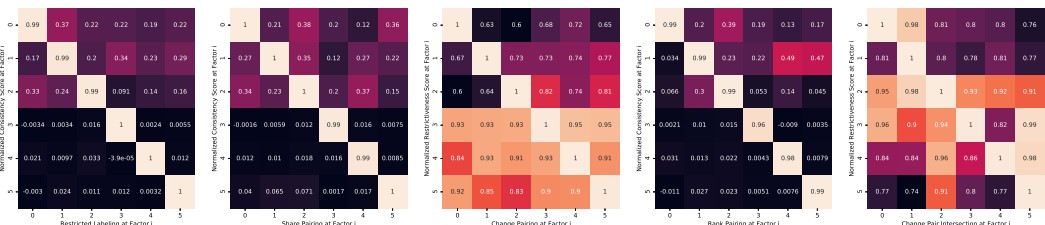

Figure 7: This is the same plot as Figure 7, but where we restrict our hyperparameter sweep to always set `extra dense = False`. See Appendix H for details about hyperparameter sweep.

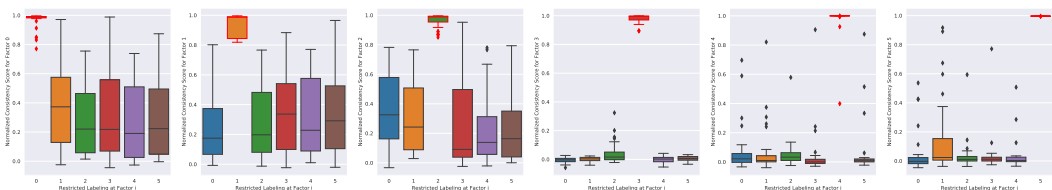

Figure 8: Restricted pairing guarantees consistency. Each plot shows the normalized consistency score of each model for each factor of variation. Our theory predicts each boxplot highlighted in red to achieve the highest consistency. Due to the prevalence of restricted pairing in the existing literature, we chose to only conduct the single-factor restricted labeling experiment on Shapes3D.

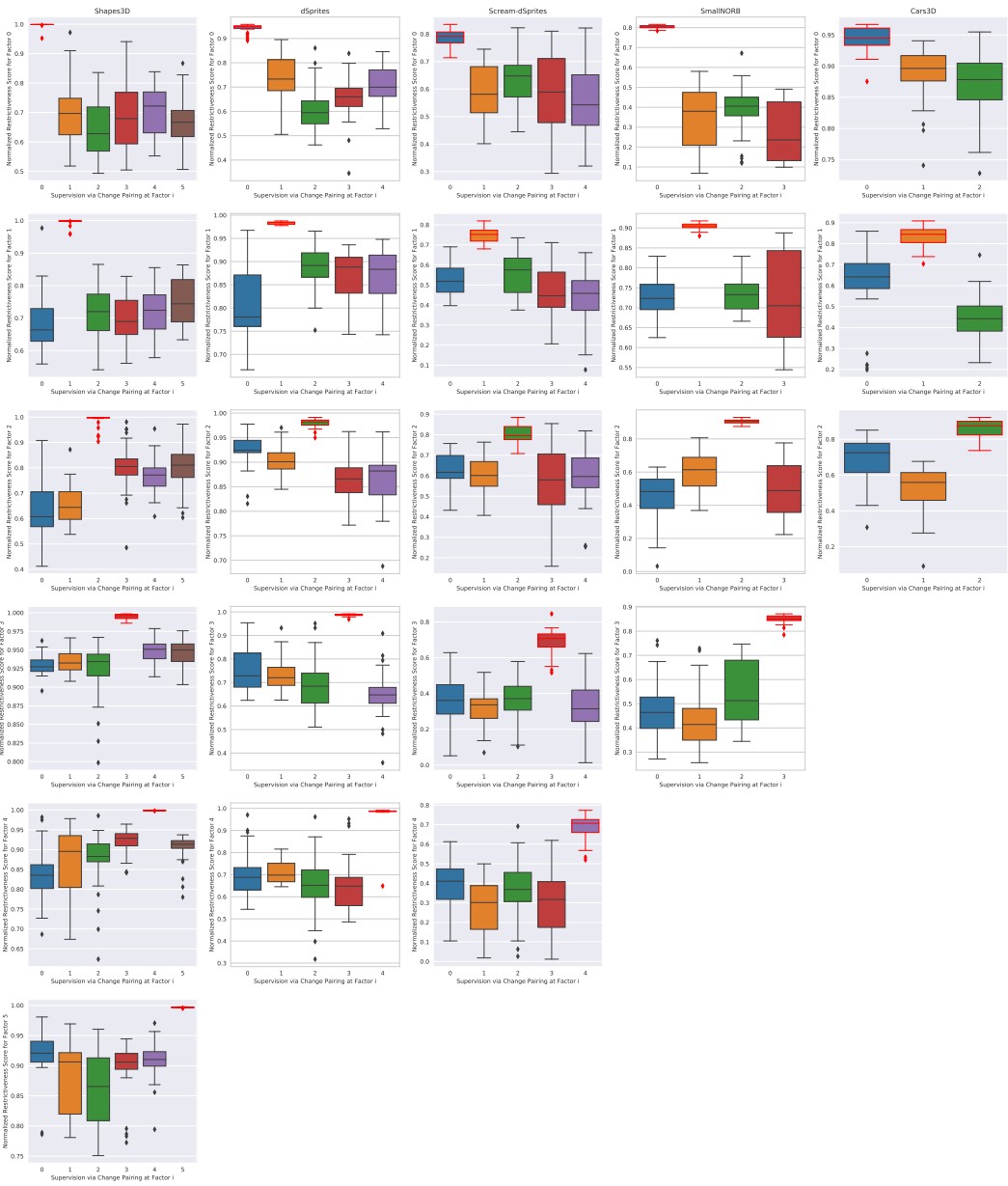

Figure 9: Change pairing guarantees restrictiveness. Each plot shows normalized restrictiveness score of each model for each factor of variation (row) across different datasets (columns). Different colors indicate models trained with change pairing on different factors. The appropriately-supervised model for each factor is marked in red.

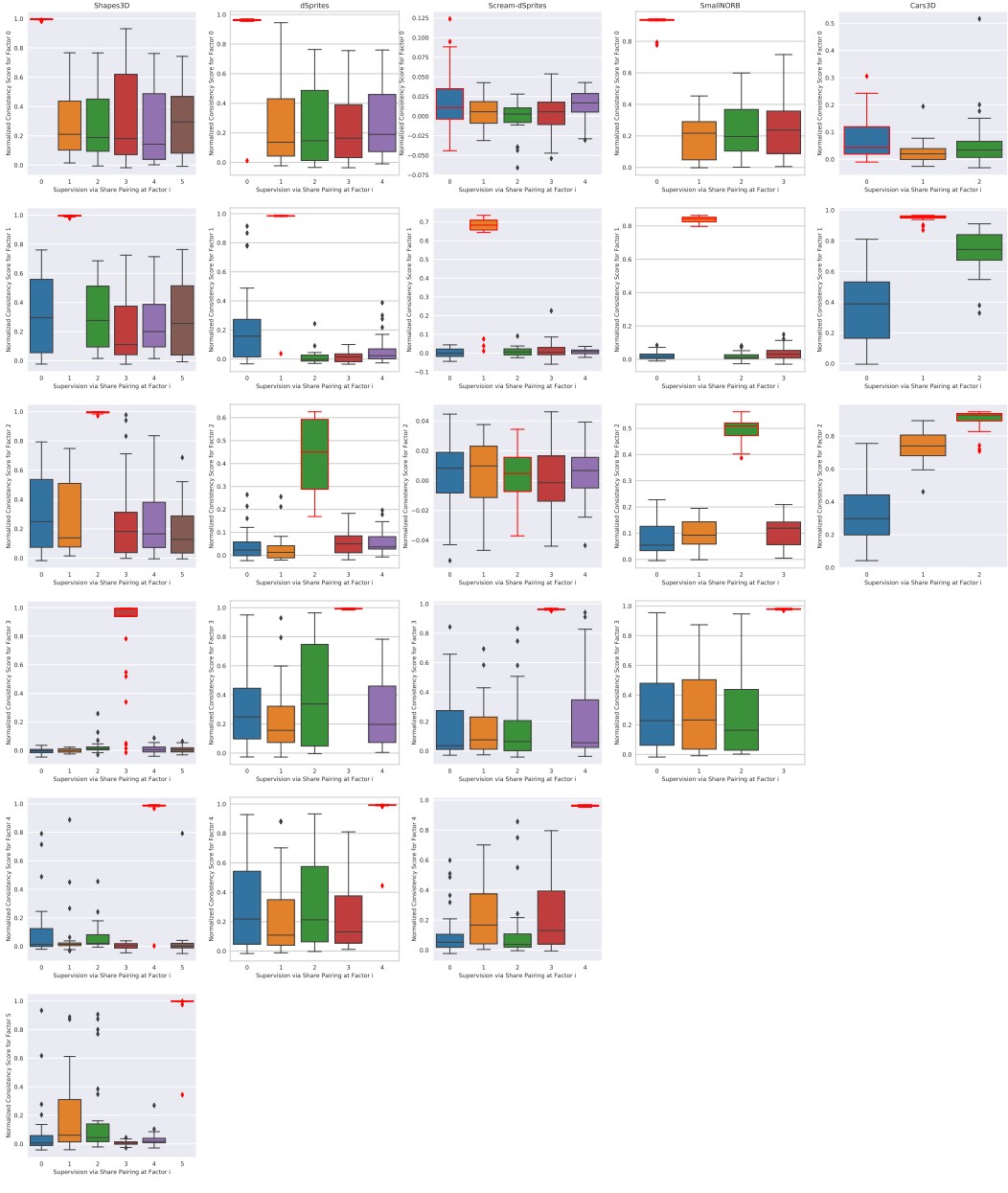

Figure 10: Share pairing guarantees consistency. Each plot shows normalized consistency score of each model for each factor of variation (row) across different datasets (columns). Different colors indicate models trained with share pairing on different factors. The appropriately-supervised model for each factor is marked in red.

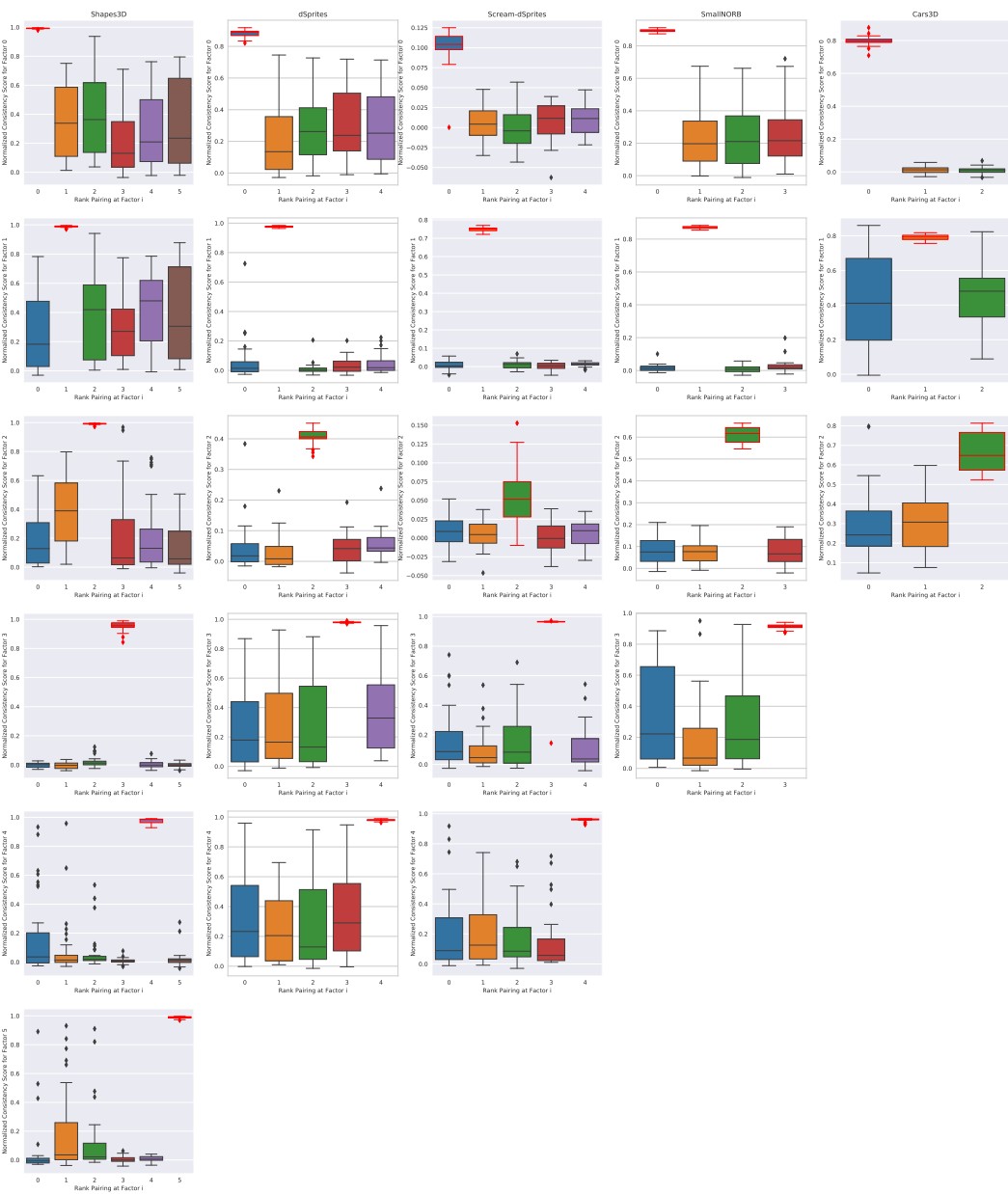

Figure 11: Rank pairing guarantees consistency. Each plot shows normalized consistency score of each model for each factor of variation (row) across different datasets (columns). Different colors indicate models trained with rank pairing on different factors. The appropriately-supervised model for each factor is marked in red.

# E  CONSISTENCY VERSUS RESTRICTIVENESS

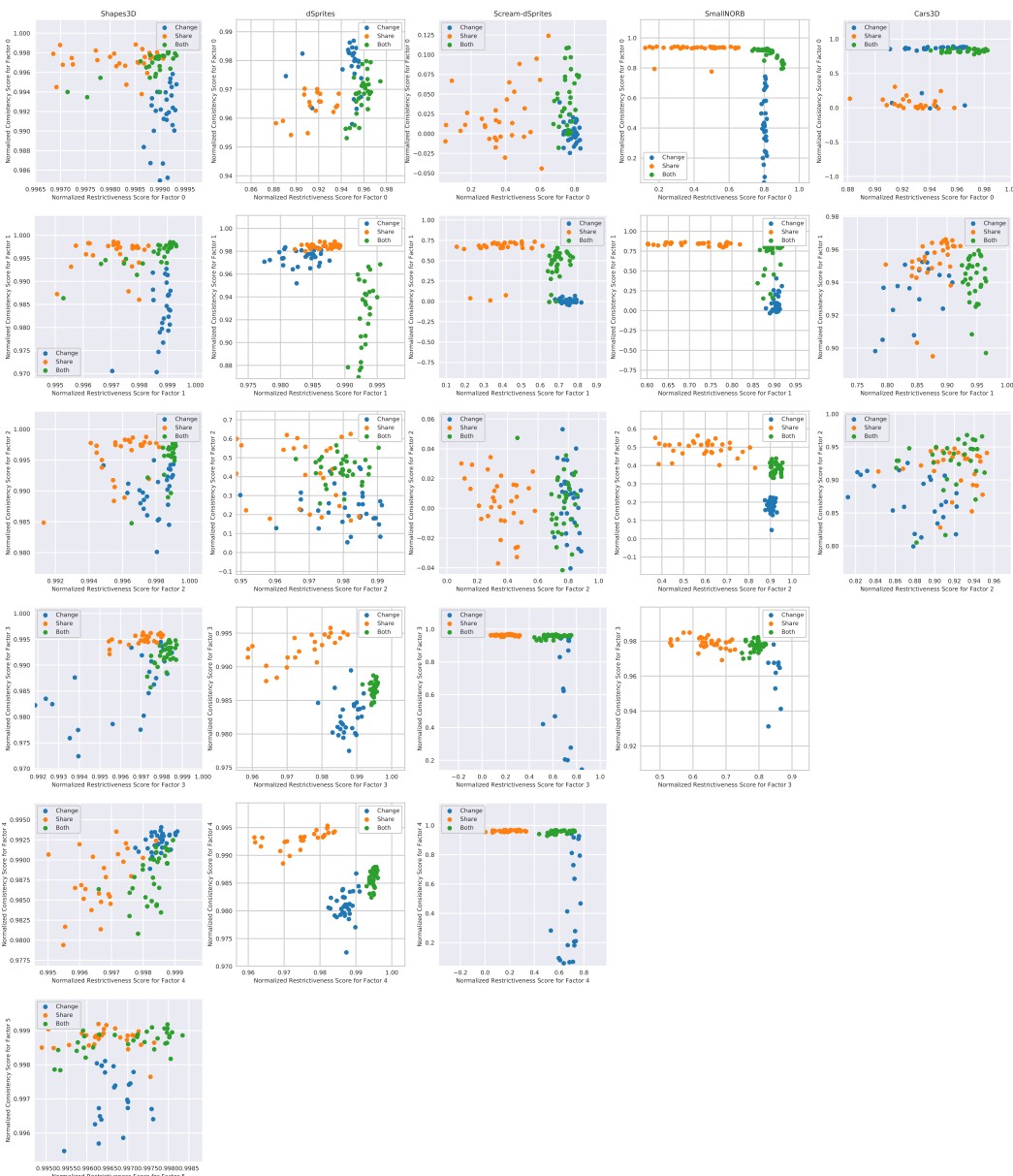

Figure 12: Normalized consistency vs. restrictiveness score of different models on each factor (row) across different datasets (columns). In many of the plots, we see that models trained via change-sharing (blue) achieve higher restrictiveness; models trained via share-sharing (orange) achieve higher consistency; models trained via both techniques (green) simultaneously achieve restrictiveness and consistency in most cases.

# F    FULL DISENTANGLEMENT EXPERIMENTS

Figure 13: Disentanglement performance of a vanilla GAN, share pairing GAN, change pairing GAN, rank pairing GAN, and fully-labeled GAN, as measured by multiple disentanglement metrics in existing literature (rows) across multiple datasets (columns). According to almost all metrics, our weakly supervised models surpass the baseline, and in some cases, even outperform the fully-labeled model.

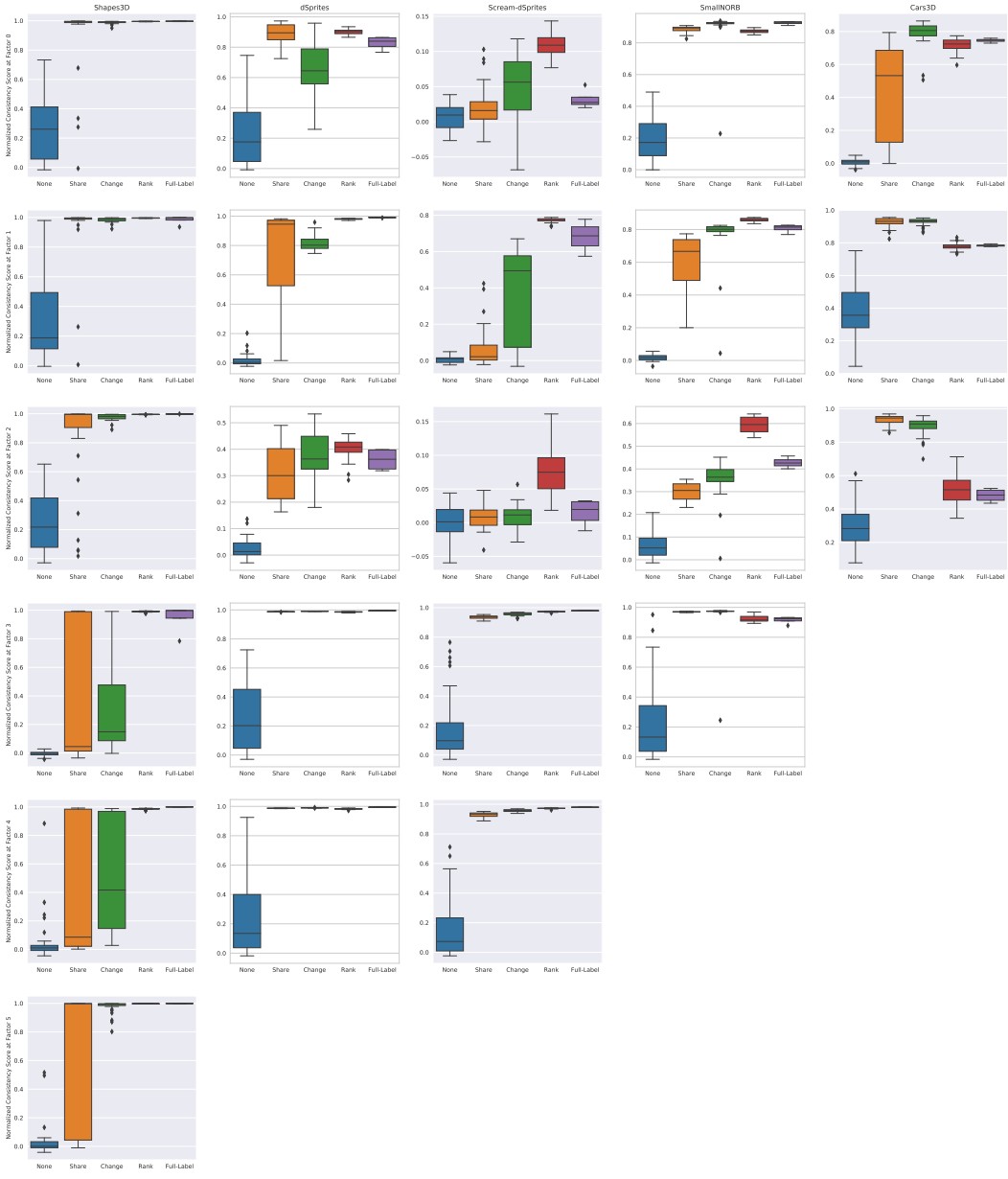

Figure 14: Performance of a vanilla GAN (blue), share pairing GAN (orange), change pairing GAN (green), rank pairing GAN (red), and fully-labeled GAN (purple), as measured by normalized consistency score of each factor (rows) across multiple datasets (columns). Factors $\{3, 4, 5\}$ in the first column shows that distribution matching to all six change / share pairing datasets is particularly challenging for the models when trained on certain hyperparameter choices. However, since consistency and restrictiveness can be measured in weakly supervised settings, it suffices to use these metrics for hyperparameter selection. We see in Figure 16 and Appendix G that using consistency and restrictiveness for hyperparameter selection serves as a viable weakly-supervised surrogate for existing fully-supervised disentanglement metrics.

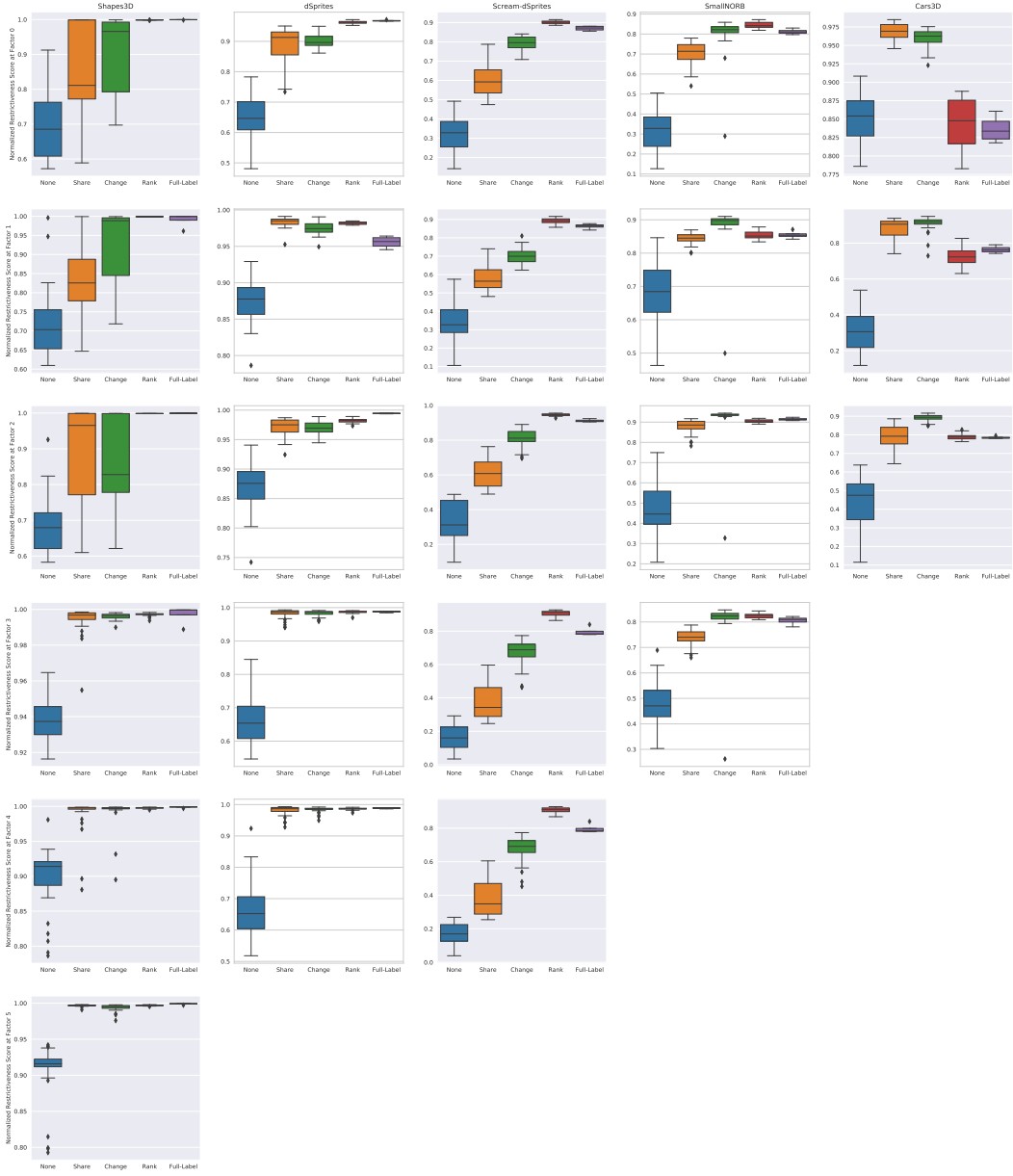

Figure 15: Performance of a vanilla GAN (blue), share pairing GAN (orange), change pairing GAN (green), rank pairing GAN (red), and fully-labeled GAN (purple), as measured by normalized restrictiveness score of each factor (rows) across multiple datasets (columns). Since restrictiveness and consistency are complementary, we see that the anomalies in Figure 14 are reflected in the complementary factors in this figure.

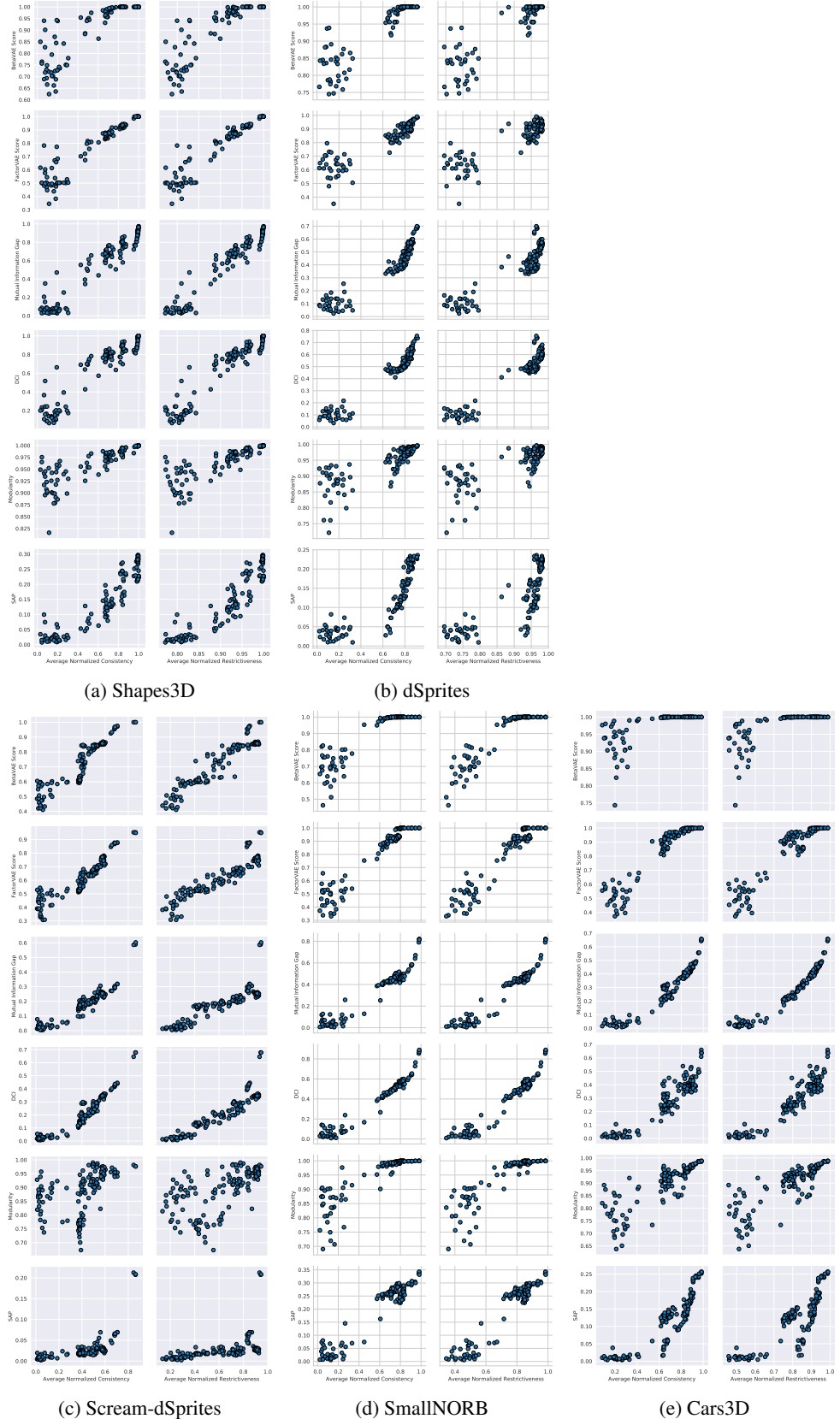

Figure 16: Scatterplot of existing disentanglement metrics versus average normalized consistency and restrictiveness. Whereas existing disentanglement metrics are fully-supervised, it is possible to measure average normalized consistency and restrictiveness with weakly supervised data (share-pairing and match-pairing respectively), making it viable to perform hyperparameter tuning under weakly supervised conditions.

## G    FULL DISENTANGLEMENT VISUALIZATIONS

As a demonstration of the weakly-supervised generative models, we visualize our best-performing match-pairing generative models (as selected according to the normalized consistency score averaged across all the factors). Recall from Figures 2a to 2c that, to visually check for consistency and restrictiveness, it is important that we not only ablate a single factor (across the column), but also show that the factor stays consistent (down the row). Each block of $3 \times 12$ images in Figures 17 to 21 checks for disentanglement of the corresponding factor. Each row is constructed by random sampling of $Z_{\setminus i}$ and then ablating $Z_i$.

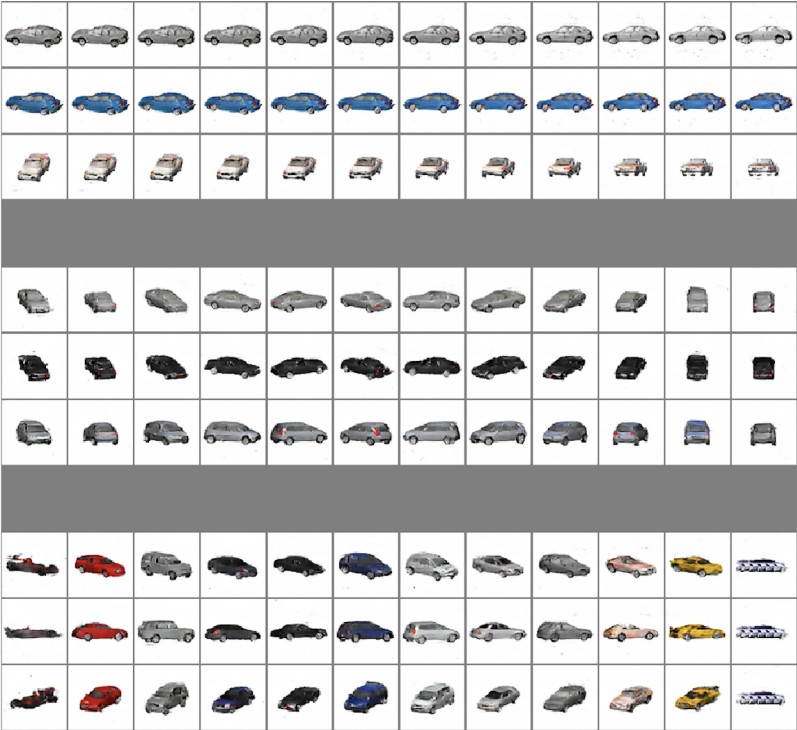

Figure 17: Cars3D. Ground truth factors: elevation, azimuth, object type.

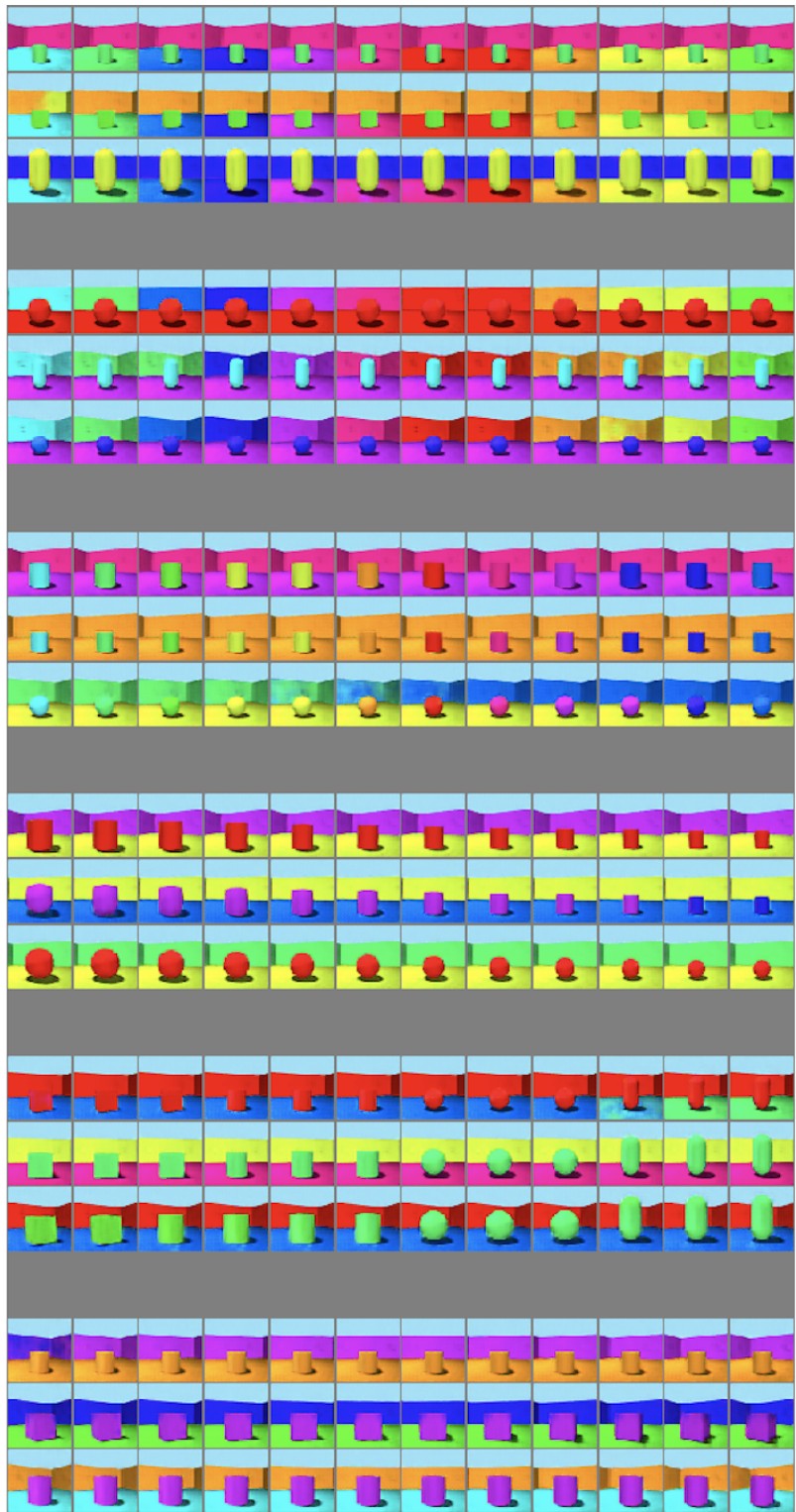

Figure 18: Shapes3D. Ground truth factors: floor color, wall color, object color, object size, object type, and azimuth.

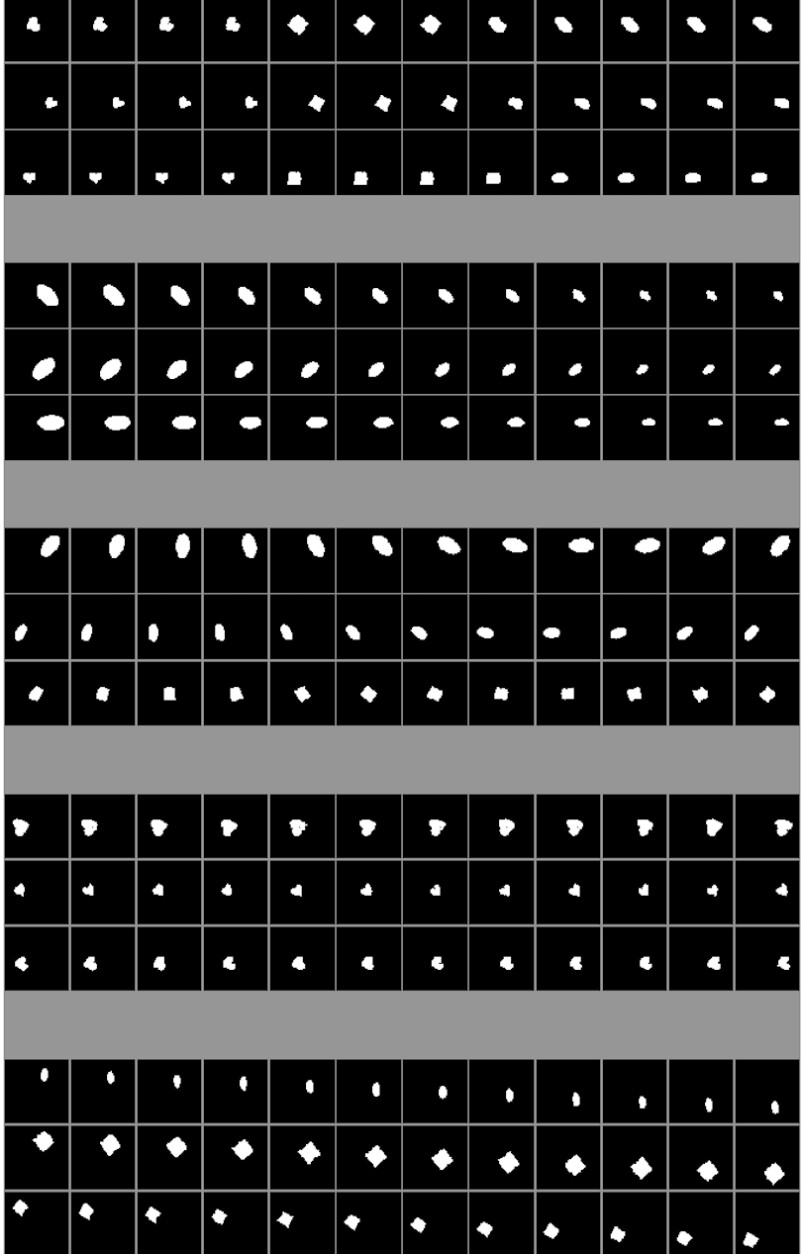

Figure 19: dSprites. Ground truth factors: shape, scale, orientation, X-position, Y-position.

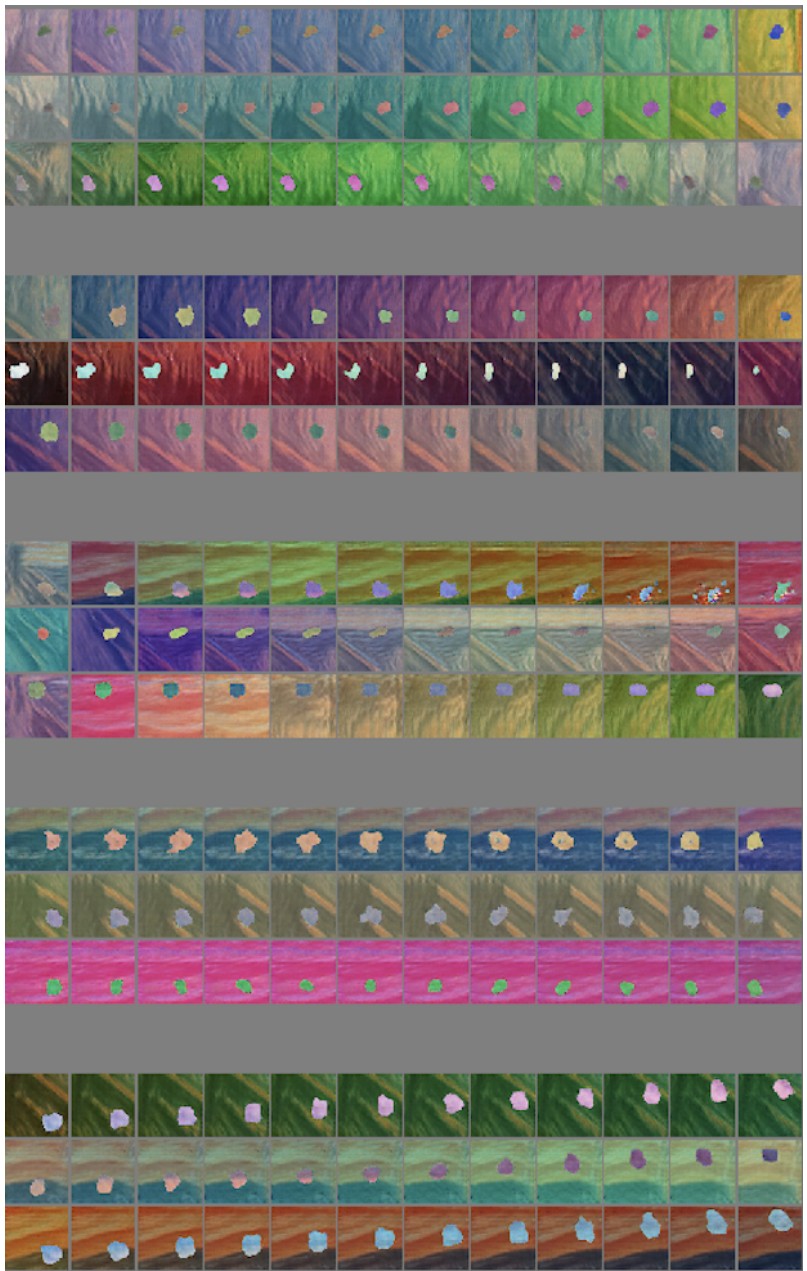

Figure 20: Scream-dSprites. Ground truth factors: shape, scale, orientation, X-position, Y-position.

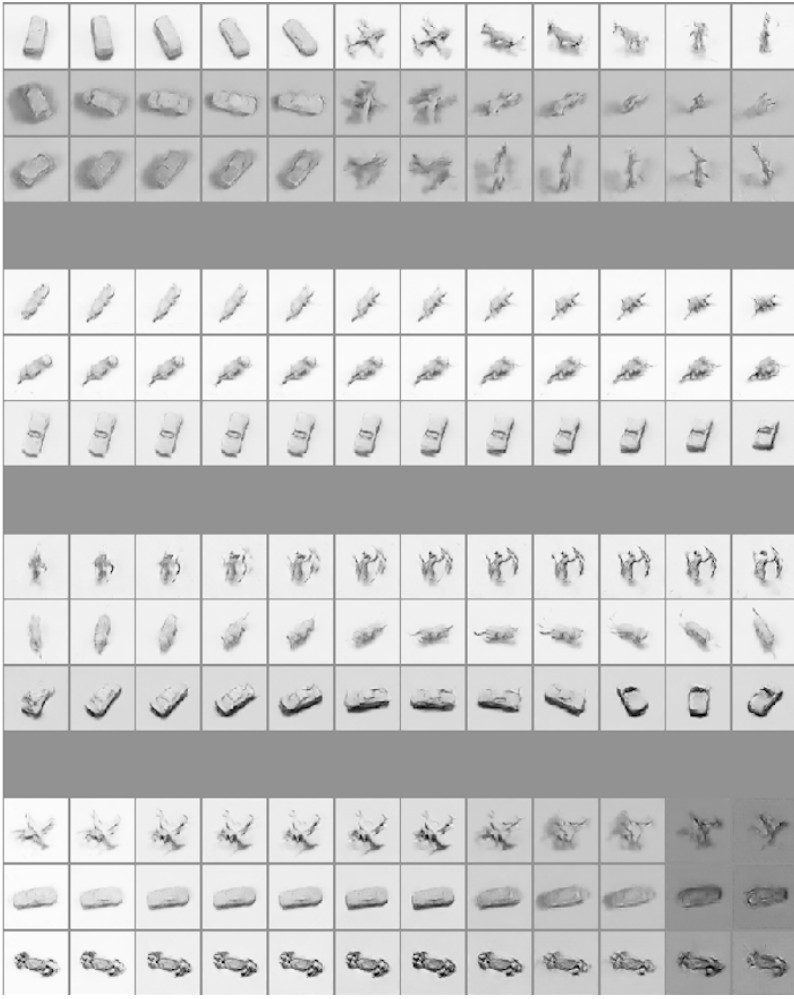

Figure 21: SmallNORB. Ground truth factors: category, elevation, azimuth, lighting condition.

# H  HYPERPARAMETERS

Table 1: We trained a probablistic Gaussian encoder to approximately invert the generative model. The encoder is not trained jointly with the generator, but instead trained separately from the generative model (i.e. encoder gradient does not backpropagate to generative model). During training, the encoder is only exposed to data generated by the learned generative model.

| Encoder |
| --- |
| $4 \times 4$ spectral norm conv. 32. lReLU |
| $4 \times 4$ spectral norm conv. 32. lReLU |
| $4 \times 4$ spectral norm conv. 64. lReLU |
| $4 \times 4$ spectral norm conv. 64. lReLU |
| flatten |
| 128 spectral norm dense. lReLU |
| $2 \times z$-dim spectral norm dense |

Table 2: Generative model architecture.

| Generator |
| --- |
| 128 dense. ReLU. batchnorm. |
| 1024 dense. ReLU. batchnorm. |
| $4 \times 4 \times 64$ reshape. |
| $4 \times 4$ conv. 64. lReLU. batchnorm. |
| $4 \times 4$ conv. 32. lReLU. batchnorm. |
| $4 \times 4$ conv. 32. lReLU. batchnorm. |
| $4 \times 4$ conv. 3. sigmoid |

Table 3: Discriminator used for restricted labeling. Parts in red are part of hyperparameter search.

| Discriminator Body |
| --- |
| $4 \times 4$ spectral norm conv. $32 \times$ width. lReLU |
| $4 \times 4$ spectral norm conv. $32 \times$ width. lReLU |
| $4 \times 4$ spectral norm conv. $64 \times$ width. lReLU |
| $4 \times 4$ spectral norm conv. $64 \times$ width. lReLU |
| flatten |
| if extra dense: $128 \times$ width spectral norm dense. lReLU |
| **Discriminator Auxiliary Channel for Label** |
| $128 \times$ width spectral norm dense. lReLU |
| If extra dense: $128 \times$ width spectral norm dense. lReLU |
| **Discriminator head** |
| concatenate body and auxiliary. |
| $128 \times$ width spectral norm dense. lReLU |
| $128 \times$ width spectral norm dense. lReLU |
| 1 spectral norm dense with bias. |

Table 4: Discriminator used for match pairing. We use a projection discriminator (Miyato & Koyama, 2018) and thus have an unconditional and conditional head. Parts in red are part of hyperparameter search.

| Discriminator Body Applied Separately to $x$ and $x'$ |
|---|
| $4 \times 4$ spectral norm conv. $32 \times$ width. lReLU |
| $4 \times 4$ spectral norm conv. $32 \times$ width. lReLU |
| $4 \times 4$ spectral norm conv. $64 \times$ width. lReLU |
| $4 \times 4$ spectral norm conv. $64 \times$ width. lReLU |
| flatten |
| If extra dense: $128 \times$ width spectral norm dense. lReLU |
| concatenate the pair. |
| $128 \times$ width spectral norm dense. lReLU |
| $128 \times$ width spectral norm dense. lReLU |
| **Unconditional Head** |
| 1 spectral norm dense with bias |
| **Conditional Head** |
| $128 \times$ width spectral norm dense |

Table 5: Discriminator used for rank pairing. For rank-pairing, we use a special variant of the projection discriminator, where the conditional logit is computed via taking the difference between the two pairs and multiplying by $y \in \{-1, +1\}$. The discriminator is thus implicitly taking on the role of an adversarially trained encoder that checks for violations of the ranking rule in the embedding space. Parts in red are part of hyperparameter search.

| Discriminator Body Applied Separately to $x$ and $x'$ |
|---|
| $4 \times 4$ spectral norm conv. $32 \times$ width. lReLU |
| $4 \times 4$ spectral norm conv. $32 \times$ width. lReLU |
| $4 \times 4$ spectral norm conv. $64 \times$ width. lReLU |
| $4 \times 4$ spectral norm conv. $64 \times$ width. lReLU |
| flatten |
| If extra dense: $128 \times$ width spectral norm dense. lReLU |
| concatenate the pair. |
| **Unconditional Head Applied Separately to $x$ and $x'$** |
| 1 spectral norm dense with bias. |
| **Conditional Head Applied Separately to $x$ and $x'$** |
| $y$-dim spectral norm dense. |

For all models, we use the Adam optimizer with $\beta_1 = 0.5, \beta_2 = 0.999$ and set the generator learning rate to $1 \times 10^{-3}$. We use a batch size of $64$ and set the leaky ReLU negative slope to $0.2$.

To demonstrate some degree of robustness to hyperparameter choices, we considered five different ablations:

1. Width multiplier on the discriminator network ($\{1, 2\}$)
2. Whether to add an extra fully-connected layer to the discriminator ($\{\text{True}, \text{False}\}$).
3. Whether to add a bias term to the head ($\{\text{True}, \text{False}\}$).
4. Whether to use two-time scale learning rate by setting encoder+discriminator learning rate multipler to ($\{1, 2\}$).
5. Whether to use the default PyTorch or Keras initialization scheme in all models.

As such, each of our experimental setting trains a total of $32$ *distinct* models. The only exception is the intersection experiments where we fixed the width multiplier to $1$.

To give a sense of the scale of our experimental setup, note that the $864$ models in Figure 4 originate as follows:

1. $32$ hyperparameter conditions $\times$ 6 restricted labeling conditions.
2. $32$ hyperparameter conditions $\times$ 6 match pairing conditions.
3. $32$ hyperparameter conditions $\times$ 6 share pairing conditions.
4. $32$ hyperparameter conditions $\times$ 6 rank pairing conditions.
5. $16$ hyperparameter conditions $\times$ 6 intersection conditions.

# I PROOFS

## I.1 ASSUMPTIONS ON $\mathcal{H}$

**Assumption 1.** Let $D \subseteq [n]$ indexes discrete random variables $S_D$. Assume that the remaining random variables $S_C = S_{\setminus D}$ have probability density function $p(s_C | s_D)$ for any set of values $s_D$ where $p(S_D = s_D) > 0$.

**Assumption 2.** Without loss of generality, suppose $S_{1:n} = [S_C, S_D]$ is ordered by concatenating the continuous variables with the discrete variables. Let $\mathcal{B}(s_D) = [\text{int}(\text{supp}(p(s_C | s_D))), s_D]$ denote the interior of the support of the continuous conditional distribution of $S_C$ concatenated with its conditioning variable $s_D$ drawn from $S_D$. With a slight abuse of notation, let $\mathcal{B}(S) = \bigcup_{s_D : p(s_D > 0)} \mathcal{B}(s_D)$. We assume $\mathcal{B}(S)$ is *zig-zag connected*, i.e., for any $I, J \subseteq [n]$, for any two points $s_{1:n}, s'_{1:n} \in \mathcal{B}(S)$ that only differ in coordinates in $I \cup J$, there exists a path $\{s^t_{1:n}\}_{t=0:T}$ contained in $\mathcal{B}(S)$ such that

$$s^0_{1:n} = s_{1:n} \tag{13}$$

$$s^T_{1:n} = s'_{1:n} \tag{14}$$

$$\forall \, 0 \le t < T, \text{ either } s^t_{\setminus I} = s^{t+1}_{\setminus I} \text{ or } s^t_{\setminus J} = s^{t+1}_{\setminus J}, \tag{15}$$

Intuitively, this assumption allows transition from $s_{1:n}$ to $s'_{1:n}$ via a series of modifications that are only in $I$ or only in $J$. Note that zig-zag connectedness is necessary for restrictiveness union (Proposition 3) and consistency intersection (Proposition 4). Fig. 22 gives examples where restrictiveness union is not satisfied when zig-zag connectedness is violated.

**Assumption 3.** For arbitrary coordinate $j \in [m]$ of $g$ that maps to a continuous variable $X_j$, we assume that $g_j(s)$ is continuous at $s$, $\forall s \in \mathcal{B}(S)$; For arbitrary coordinate $j \in [m]$ of $g$ that maps to a discrete variable $X_j$, $\forall s_D$ where $p(s_D) > 0$, we assume that $g_j(s)$ is constant over each connected component of $\text{int}(\text{supp}(p(s_C | s_D)))$.

Define $\mathcal{B}(X)$ analogously to $\mathcal{B}(S)$. Symmetrically, for arbitrary coordinate $i \in [n]$ of $e$ that maps to a continuous variable $S_i$, assume that $e_i(x)$ is continuous at $x$, $\forall x \in \mathcal{B}(X)$; For arbitrary coordinate $i \in [n]$ of $e$ that maps to a discrete $S_i$, $\forall x_D$ where $p(x_D) > 0$, we assume that $e_i(x)$ is constant over each connected component of $\text{int}(\text{supp}(p(x_C | x_D)))$.

**Assumption 4.** Assume that every factor of variation is recoverable from the observation $\mathcal{X}$. Formally, $(p, g, e)$ satisfies the following property

$$\mathbb{E}_{p(s_{1:n})} \| e \circ g(s_{1:n}) - s_{1:n} \|^2 = 0. \tag{16}$$

## I.2 CALCULUS OF DISENTANGLEMENT

### I.2.1 EXPECTED-NORM REDUCTION LEMMA

**Lemma 1.** *Let $x, y$ be two random variables with distribution $p$, $f(x, y)$ be arbitrary function. Then*

$$\mathbb{E}_{x \sim p(x)} \mathbb{E}_{y, y' \sim p(y|x)} \| f(x, y) - f(x, y') \|^2 \le \mathbb{E}_{(x,y),(x',y') \sim p(x,y)} \| f(x, y) - f(x', y') \|^2.$$

*Proof.* Assume w.l.o.g that $\mathbb{E}_{(x,y) \sim p(x,y)} f(x, y) = 0$.

$$LHS = 2\mathbb{E}_{(x,y) \sim p(x,y)} \| f(x, y) \|^2 - 2\mathbb{E}_{x \sim p(x)} \mathbb{E}_{y, y' \sim p(y|x)} f(x, y)^T f(x, y') \tag{17}$$

$$= 2\mathbb{E}_{(x,y) \sim p(x,y)} \| f(x, y) \|^2 - 2\mathbb{E}_{x \sim p(x)} \mathbb{E}_{y \sim p(y|x)} f(x, y)^T \mathbb{E}_{y' \sim p(y|x)} f(x, y') \tag{18}$$

$$= 2\mathbb{E}_{(x,y) \sim p(x,y)} \| f(x, y) \|^2 - 2\mathbb{E}_{x \sim p(x)} \| \mathbb{E}_{y \sim p(y|x)} f(x, y) \|^2 \tag{19}$$

$$\le 2\mathbb{E}_{(x,y) \sim p(x,y)} \| f(x, y) \|^2 \tag{20}$$

$$= 2\mathbb{E}_{(x,y) \sim p(x,y)} \| f(x, y) \|^2 - 2\| \mathbb{E}_{(x,y) \sim p(x,y)} f(x, y) \|^2 \tag{21}$$

$$= 2\mathbb{E}_{(x,y) \sim p(x,y)} \| f(x, y) \|^2 - 2\mathbb{E}_{(x,y),(x',y') \sim p(x,y)} f(x, y)^T f(x', y') \tag{22}$$

$$= RHS. \tag{23}$$

$\square$

### I.2.2 CONSISTENCY UNION

Let $L = I \cap J, K = \backslash (I \cup J), M = I - L, N = J - L.$

**Proposition 2.** $C(I) \wedge C(J) \implies C(I \cup J).$

*Proof.*

$$C(I) \implies \mathbb{E}_{z_M, z_L} \mathbb{E}_{z_N, z'_N, z_K, z'_K} \| r_I \circ G(z_M, z_L, z_N, z_K) - r_I \circ G(z_M, z_L, z'_N, z'_K) \|^2 = 0. \tag{24}$$

For any fixed value of $z_M, z_L,$

$$\mathbb{E}_{z_N, z'_N, z_K, z'_K} \| r_I \circ G(z_M, z_L, z_N, z_K) - r_I \circ G(z_M, z_L, z'_N, z'_K) \|^2 \tag{25}$$

$$\geq \mathbb{E}_{z_N} \mathbb{E}_{z_K, z'_K} \| r_I \circ G(z_M, z_L, z_N, z_K) - r_I \circ G(z_M, z_L, z_N, z'_K) \|^2. \tag{26}$$

by plugging in $x = z_N, y = z_K$ into Lemma 1. Therefore

$$C(I) \implies \mathbb{E}_{z_M, z_L, z_N} \mathbb{E}_{z_K, z'_K} \| r_I \circ G(z_M, z_L, z_N, z_K) - r_I \circ G(z_M, z_L, z_N, z'_K) \|^2 = 0. \tag{27}$$

Similarly we have

$$C(J) \implies \mathbb{E}_{z_M, z_L, z_N} \mathbb{E}_{z_K, z'_K} \| r_J \circ G(z_M, z_L, z_N, z_K) - r_J \circ G(z_M, z_L, z_N, z'_K) \|^2 = 0 \tag{28}$$

$$\implies \mathbb{E}_{z_M, z_L, z_N} \mathbb{E}_{z_K, z'_K} \| r_N \circ G(z_M, z_L, z_N, z_K) - r_N \circ G(z_M, z_L, z_N, z'_K) \|^2 = 0. \tag{29}$$

As $I \cap N = \emptyset, I \cup N = I \cup J$, adding the above two equations gives us $C(I \cup J)$. $\square$

### I.2.3 RESTRICTIVENESS UNION

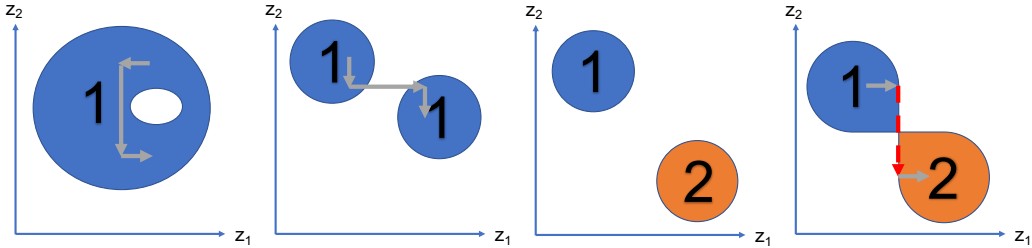

Figure 22: Zig-zag connectedness is necessary for restriveness union. Here $n = m = 3$. Colored areas indicate the support of $p(z_1, z_2)$; the marked numbers indicate the measurement of $s_3$ given $(z_1, z_2)$. Left two panels satisfy zig-zag connectedness (the paths are marked in gray) while the right two do not (indeed $R(1) \wedge R(2) \nRightarrow R(\{1, 2\})$). In the right-most panel, any zig-zag path connecting two points from blue and orange areas has to pass through boundary of the support (disallowed).

Similarly define index sets $L, K, M, N$.

**Proposition 3.** Under assumptions specified in Appendix I.1, $R(I) \wedge R(J) \implies R(I \cup J).$

*Proof.* Denote $f = e_K^* \circ g$. We claim that

$$R(I) \iff \mathbb{E}_{z_{\backslash I}} \mathbb{E}_{z_I, z'_I} \| f(z_I, z_{\backslash I}) - f(z'_I, z_{\backslash I}) \|^2 = 0. \tag{30}$$

$$\iff \forall (z_I, z_{\backslash I}), (z'_I, z_{\backslash I}) \in \mathcal{B}(Z), f(z_I, z_{\backslash I}) = f(z'_I, z_{\backslash I}). \tag{31}$$

We first prove the backward direction: When we draw $z_{\backslash I} \sim p(z_{\backslash I}), z_I, z'_I \sim p(z_I | z_{\backslash I})$, let $E_1$ denote the event that $(z_I, z_{\backslash I}) \notin \mathcal{B}(Z)$, and $E_2$ denote the event that $(z'_I, z_{\backslash I}) \notin \mathcal{B}(Z)$. Reorder the indices of $(z_I, z_{\backslash I})$ as $(z_C, z_D)$. The probability that $(z_I, z_{\backslash I}) \notin \mathcal{B}(Z)$ (i.e.,$z_C$ is on the boundary of $\mathcal{B}(z_D)$) is 0. Therefore $\Pr[E_1] = \Pr[E_2] = 0$. Therefore $\Pr[E_1 \cup E_2] \leq \Pr[E_1] + \Pr[E_2] = 0$, i.e., with probability 1, $\| f(z_I, z_{\backslash I}) - f(z'_I, z_{\backslash I}) \|^2 = 0$.

Now we prove the forward direction: Assume for the sake of contradiction that $\exists (z_I, z_{\setminus I}), (z'_I, z_{\setminus I}) \in \mathcal{B}(Z)$ such that $f(z_I, z_{\setminus I}) < f(z'_I, z_{\setminus I})$. Denote $U = I \cap D$, $V = I \cap C$, $W = \setminus I \cap D$, $Q = \setminus I \cap C$. We have $f(z_U, z_V, z_W, z_Q) < f(z'_U, z'_V, z_W, z_Q)$. Since $f$ is continuous (or constant) at $(z_U, z_V, z_W, z_Q)$ in the interior of $\mathcal{B}([z_U, z_W])$, and $f$ is also continuous (or constant) at $(z'_U, z'_V, z_W, z_Q)$ in the interior of $\mathcal{B}([z'_U, z_W])$, we can draw open balls of radius $r > 0$ around each point, i.e., $B_r(z_V, z_Q) \subset \mathcal{B}([z_U, z_W])$ and $B_r(z'_V, z_Q) \subset \mathcal{B}([z'_U, z_W])$, where

$$\forall (z_V^*, z_Q^*) \in B_r(z_V, z_Q), \forall (z_V^\Delta, z_Q^\Delta) \in B_r(z'_V, z_Q), f(z_U, z_V^*, z_W, z_Q^*) < f(z'_U, z_V^\Delta, z_W, z_Q^\Delta). \tag{32}$$

When we draw $z_{\setminus I} \sim p(z_{\setminus I}), z_I, z'_I \sim p(z_I | z_{\setminus I})$, let $C$ denote the event that $(z_I, z_{\setminus I}) = (z_V^*, z_U, z_Q^\#, z_W)$, $(z'_I, z_{\setminus I}) = (z_V^\Delta, z'_U, z_Q^\#, z_W)$ where $(z_V^*, z_Q^\#) \in B_r(z_V, z_Q)$ and $(z_V^\Delta, z_Q^\#) \in B_r(z'_V, z_Q)$. Since both balls have positive volume, $\Pr[C] > 0$. However, $\|f(z_I, z_{\setminus I}) - f(z'_I, z_{\setminus I})\|^2 > 0$ whenever event $C$ happens, which contradicts $R(I)$. Therefore $\forall (z_I, z_{\setminus I}), (z'_I, z_{\setminus I}) \in \mathcal{B}(Z), f(z_I, z_{\setminus I}) = f(z'_I, z_{\setminus I})$.

We have shown that

$$R(I) \iff \forall (z_M, z_L, z_N, z_K), (z'_M, z'_L, z_N, z_K) \in \mathcal{B}(Z), f(z_M, z_L, z_N, z_K) = f(z'_M, z'_L, z_N, z_K). \tag{33}$$

Similarly

$$R(J) \iff \forall (z_M, z_L, z_N, z_K), (z_M, z'_L, z'_N, z_K) \in \mathcal{B}(Z), f(z_M, z_L, z_N, z_K) = f(z_M, z'_L, z'_N, z_K). \tag{34}$$

$$R(I \cup J) \iff \forall (z_M, z_L, z_N, z_K), (z'_M, z'_L, z'_N, z_K) \in \mathcal{B}(Z), f(z_M, z_L, z_N, z_K) = f(z'_M, z'_L, z'_N, z_K) \tag{35}$$

Let the zig-zag path between $(z_M, z_L, z_N, z_K)$ and $(z'_M, z'_L, z'_N, z_K) \in \mathcal{B}(Z)$ be $\{(z_M^t, z_L^t, z_N^t, z_K)\}_{t=0}^T$. Repeatedly applying the equivalent conditions of $R(I)$ and $R(J)$ gives us

$$f(z_M, z_L, z_N, z_K) = f(z_M^1, z_L^1, z_N^1, z_K) = \cdots = f(z_M^{T-1}, z_L^{T-1}, z_N^{T-1}, z_K) = f(z'_M, z'_L, z'_N, z_K). \tag{36}$$

$\square$

### I.3 CONSISTENCY AND RESTIVENESS INTERSECTION

**Proposition 4.** Under the same assumptions as restrictiveness union, $C(I) \wedge C(J) \implies C(I \cap J)$.

*Proof.*

$$C(I) \wedge C(J) \implies R(\setminus I) \wedge R(\setminus J) \tag{37}$$
$$\implies R(\setminus I \cup \setminus J) \tag{38}$$
$$\implies C(\setminus(\setminus I \cup \setminus J)) \tag{39}$$
$$\implies C(I \cap J). \tag{40}$$

$\square$

**Proposition 5.** $R(I) \wedge R(J) \implies R(I \cap J)$.

Proof is analogous to Proposition 4.

### I.4 DISTRIBUTION MATCHING GUARANTEES LATENT CODE INFORMATIVENESS

**Proposition 6.** If $(p^*, g^*, e^*) \in \mathcal{H}$, and $(p, g, e) \in \mathcal{H}$, and $g^*(S) \overset{d}{=} g(Z)$, then there exists a continuous function $r$ such that

$$\mathbb{E}_{p(s_{1:n})} \|r \circ e \circ g^*(s) - s\| = 0. \tag{41}$$

*Proof.* We show that $r = e^* \circ g$ satisfies Proposition 6. By Assumption 4,

$$\mathbb{E}_s \| e^* \circ g^*(s) - s \|^2 = 0. \tag{42}$$

$$\mathbb{E}_z \| e \circ g(z) - z \|^2 = 0. \tag{43}$$

By the same reasoning as in the proof of Proposition 3,

$$\mathbb{E}_s \| e^* \circ g^*(s) - s \|^2 = 0 \implies \forall s \in \mathcal{B}(S), e^* \circ g^*(s) = s. \tag{44}$$

$$\mathbb{E}_z \| e \circ g(z) - z \|^2 = 0 \implies \forall z \in \mathcal{B}(Z), e \circ g(z) = z. \tag{45}$$

Let $s \sim p(s)$. We claim that $\Pr[E_1] = 1$, where $E_1$ denote the event that $\exists z \in \mathcal{B}(Z)$ such that $g^*(s) = g(z)$. Suppose to the contrary that there is a measure-non-zero set $\mathcal{S} \subseteq supp(p(s))$ such that $\forall s \in \mathcal{S}$, no $z \in \mathcal{B}(Z)$ satisfies $g^*(s) = g(z)$. Let $\mathcal{X} = \{g(s) : s \in \mathcal{S}\}$. As $g^*(S) \stackrel{d}{=} g(Z)$, $\Pr_s[g^*(s) \in \mathcal{X}] = \Pr_z[g(z) \in \mathcal{X}] > 0$. Therefore $\exists \mathcal{Z} \subseteq supp(p(z)) - \mathcal{B}(Z)$ such that $\mathcal{X} \subseteq \{g(z) : z \in \mathcal{Z}\}$. But $supp(p(z)) - \mathcal{B}(Z)$ has measure 0. Contradiction.

When we draw $s$, let $E_2$ denote the event that $s \in \mathcal{B}(S)$. $\Pr[E_2] = 1$, so $\Pr[E_1 \wedge E_2] = 1$. When $E_1 \wedge E_2$ happens, $e^* \circ g \circ e \circ g^*(s) = e^* \circ g \circ e \circ g(z) = e^* \circ g(z) = e^* \circ g^*(s) = s$. Therefore

$$\mathbb{E}_s \| e^* \circ g \circ e \circ g^*(s) - s \| = 0. \tag{46}$$

$\square$

## I.5 WEAK SUPERVISION GUARANTEE

**Theorem 1.** *Given any oracle $(p^*(s), g^*, e^*) \in \mathcal{H}$, consider the distribution-matching algorithm $\mathcal{A}$ that selects a model $(p(z), g, e) \in \mathcal{H}$ such that:*

1. $(g^*(S), S_I) \stackrel{d}{=} (g(Z), Z_I)$ (**Restricted Labeling**); or

2. $\left( g^*(S_I, S_{\backslash I}), g^*(S_I, S'_{\backslash I}) \right) \stackrel{d}{=} \left( g(Z_I, Z_{\backslash I}), g(Z_I, Z'_{\backslash I}) \right)$ (**Match Pairing**); or

3. $(g^*(S), g^*(S'), \mathbf{1}\{S_I \le S'_I\}) \stackrel{d}{=} (g(Z), g(Z'), \mathbf{1}\{Z_I \le Z'_I\})$ (**Rank Pairing**).

*Then $(p, g)$ satisfies $C(I; p, g, e^*)$ and $e$ satisfies $C(I; p^*, g^*, e)$.*

*Proof.* We prove the three cases separately:

1. Since $(x_d, s_I) \stackrel{d}{=} (x_g, z_I)$, consider the measurable function

$$f(a, b) = \| e^*_I(a) - b \|^2. \tag{47}$$

   We have

$$\mathbb{E} \| e^*_I(x_d) - s_I \|^2 = \mathbb{E} \| e^*_I(x_g) - z_I \|^2 = 0. \tag{48}$$

   By the same reasoning as in the proof of Proposition 3,

$$\mathbb{E}_z \| e^*_I \circ g(z) - z_I \|^2 = 0 \implies \forall z \in \mathcal{B}(Z), e^*_I \circ g(z) = z_I. \tag{49}$$

   Therefore

$$\mathbb{E}_{z_I} \mathbb{E}_{z_{\backslash I}, z'_{\backslash I}} \| e^*_I \circ g(z_I, z_{\backslash I}) - e^*_I \circ g(z_I, z'_{\backslash I}) \|^2 = 0. \tag{50}$$

   i.e., $g$ satisfies $C(I; p, g, e^*)$. By symmetry, $e$ satisfies $C(I; p^*, g^*, e)$.

2. 

$$\left( g^*(S_I, S_{\backslash I}), g^*(S_I, S'_{\backslash I}) \right) \stackrel{d}{=} \left( g(Z_I, Z_{\backslash I}), g(Z_I, Z'_{\backslash I}) \right) \tag{51}$$

$$\implies \| e^*_I \circ g^*(S_I, S_{\backslash I}) - e^*_I \circ g^*(S_I, S'_{\backslash I}) \|^2 \stackrel{d}{=} \| e^*_I \circ g(Z_I, Z_{\backslash I}) - e^*_I \circ g(Z_I, Z'_{\backslash I}) \|^2 \tag{52}$$

$$\implies \mathbb{E}_{z_I} \mathbb{E}_{z_{\backslash I}, z'_{\backslash I}} \| e^*_I \circ g(z_I, z_{\backslash I}) - e^*_I \circ g(z_I, z'_{\backslash I}) \|^2 = 0. \tag{53}$$

   So $g$ satisfies $C(I; p, g, e^*)$. By symmetry, $e$ satisfies $C(I; p^*, g^*, e)$.

3. Let $I = \{i\}$, $f = e_I^* \circ g$. Distribution matching implies that, with probability 1 over random draws of $Z$, $Z'$, the following event $P$ happens:

$$Z_I <= Z_I' \implies f(Z) <= f(Z'). \tag{54}$$

i.e.,

$$\mathbb{E}_{z,z'} \mathbf{1}[\neg P] = 0. \tag{55}$$

Let $W = \backslash I \cap D$, $Q = \backslash I \cap \backslash D$. We showed in the proof of Proposition 3 that

$$C(I) \iff \forall (z_I, z_W, z_Q), (z_I, z_W', z_Q') \in \mathcal{B}(Z), f(z_I, z_W, z_Q) = f(z_I, z_W', z_Q'). \tag{56}$$

We prove by contradiction. Suppose $\exists (z_I, z_W, z_Q), (z_I, z_W', z_Q') \in \mathcal{B}(Z)$ such that $f(z_I, z_W, z_Q) < f(z_I, z_W', z_Q')$.

(a) Case 1: $Z_I$ is discrete.
Since $f$ is constant both at $(z_I, z_W, z_Q)$ in the interior of $\mathcal{B}([z_I, z_W])$, and at $(z_I, z_W', z_Q')$ in the interior of $\mathcal{B}([z_I, z_W'])$, we can draw open balls of radius $r > 0$ around each point, i.e., $B_r(z_Q) \subset \mathcal{B}([z_I, z_W])$ and $B_r(z_Q') \subset \mathcal{B}([z_I, z_W'])$, where

$$\forall z_Q^* \in B_r(z_Q), \forall z_Q^\Delta \in B_r(z_Q'), f(z_I, z_W, z_Q^*) < f(z_I, z_W', z_Q^\Delta). \tag{57}$$

When we draw $z, z' \sim p(z)$, let $C$ denote the event that this specific value of $z_I$ is picked for both $z, z'$, and we picked $z_{\backslash I} \in B_r(z_Q')$, $z'_{\backslash I} \in B_r(z_Q)$. Since both balls have positive volume, $\Pr[C] > 0$. However, $P$ does not happen whenever event $C$ happens, since $z_I = z_I'$ but $f(z) > f(z')$, which contradicts $\Pr[P] = 1$.

(b) Case 2: $z_I$ is continuous.
Similar to case 1, we can draw open balls of radius $r > 0$ around each point, i.e., $B_r(z_I, z_Q) \subset \mathcal{B}(z_W)$ and $B_r(z_I, z_Q') \subset \mathcal{B}(z_W')$, where

$$\forall (z_I^*, z_Q^*) \in B_r(z_I, z_Q), \forall (z_I^\Delta, z_Q^\Delta) \in B_r(z_I, z_Q'), f(z_I^*, z_W, z_Q^*) < f(z_I^\Delta, z_W', z_Q^\Delta). \tag{58}$$

Let $H^1 = \{(z_I^*, z_Q^*) \in B_r(z_I, z_Q) : z_I^* >= z_I\}$, $H^2 = \{(z_I^\Delta, z_Q^\Delta) \in B_r(z_I, z_Q') : z_I^\Delta <= z_I\}$. When we draw $z, z' \sim p(z)$, let $C$ denote the event that we picked $z' \in H^1 \times \{z_W\}$, $z \in H^2 \times \{z_W'\}$. Since $H^1, H^2$ have positive volume, $\Pr[C] > 0$. However, $P$ does not happen whenever event $C$ happens, since $z_I <= z_I'$ but $f(z) > f(z')$, which contradicts $\Pr[P] = 1$.

Therefore we showed

$$\forall (z_I, z_W, z_Q), (z_I, z_W', z_Q') \in \mathcal{B}(Z), f(z_I, z_W, z_Q) = f(z_I, z_W', z_Q'), \tag{59}$$

i.e., $g$ satisfies $C(I; p, g, e^*)$. By symmetry, $e$ satisfies $C(I; p^*, g^*, e)$.

$\square$

## I.6 WEAK SUPERVISION IMPOSSIBILITY RESULT

**Theorem 2.** *Weak supervision via restricted labeling, match pairing, or ranking on $s_I$ is not sufficient for learning a generative model whose latent code $Z_I$ is restricted to $S_I$.*

*Proof.* We construct the following counterexample. Let $n = m = 3$ and $I = \{1\}$. The data generation process is $s_1 \sim \text{unif}([0, 2\pi])$, $(s_2, s_3) \sim \text{unif}(\{(x, y) : x^2 + y^2 \leq 1\})$, $g^*(s) = [s_1, s_2, s_3]$. Consider a generator $z_1 \sim \text{unif}([0, 2\pi])$, $(s_2, s_3) \sim \text{unif}(\{(x, y) : x^2 + y^2 \leq 1\})$, $g(z) = [z_1, \cos(z_1)z_2 - \sin(z_1)z_3, \sin(z_1)z_2 + \cos(z_1)z_3]$. Then $(x_d, s_I) \overset{d}{=} (x_g, z_I)$ but $R(I; p, g, e^*)$ and $R(I; p^*, g^*, e)$ does not hold. The same counterexample is applicable for match pairing and rank pairing.
$\square$

