# OpenReview forum: "Weakly Supervised Disentanglement with Guarantees"
_ICLR.cc/2020/Conference — Accept (Poster)_

### Official Review · AnonReviewer3 · 2019-10-23
**Official Blind Review #3**

**Rating:** 3

**Review:**

Summary

The paper tries to construct a theoretical framework to rigorously analyze the disentanglement guarantees of weak supervision algorithms. In particular, it focuses on two concepts, consistency and restrictiveness which provides a formalism that precisely distinguishes when disentanglement arises from supervision versus model inductive bias.

Strengths

The framework uses two simple concepts, consistency and restrictiveness for both generator and decoder. It also gives rise to a calculus. It is very useful to demonstrate the conditions under which various supervision strategies guarantee disentanglement.

The paper also did a good job clarifying how consistency and restrictiveness differ from other disentanglement concepts used in the literature.

Weaknesses

The paper does not propose effective methods for disentanglement in the weak supervision setting.

The experimental section uses very toy datasets. It is not clear how the weak supervision signal can come from in realistic applications.

**Experience Assessment:**

I have published one or two papers in this area.

**Review Assessment: Checking Correctness Of Derivations And Theory:**

I carefully checked the derivations and theory.

**Review Assessment: Checking Correctness Of Experiments:**

I carefully checked the experiments.

**Review Assessment: Thoroughness In Paper Reading:**

I read the paper thoroughly.

---

> ### Author Response · Authors · 2019-11-15
> **Response**
>
> Dear Reviewer,
>
> Thank you for acknowledging the significance of our theoretical framework. The goal of this paper is to clarify some confusion regarding the concept of disentanglement in the current literature, and to build up a theoretical framework that clarifies the guarantees actually conferred by several popular weak supervision methods. And we hope that you will re-assess the merits of our paper with this perspective in mind.
>
>
> On the Significance of our Theoretical Analysis
> -------------------------------------------------------------
> First, we would like to emphasize the inherent value of studying the theoretical guarantees of various weak supervision methods. Theorem 2 shows that existing approaches such as restricted-labeling in style-content disentanglement literature do not actually provide disentanglement guarantees. In contrast, employing our calculus and Theorem 1 allows us to design weak supervision methods that guarantee disentanglement. We believe our demonstration is of inherent value, as it exemplifies the theoretical analysis of guarantees that have so far eluded the disentanglement community. In light of the findings in Locatello et al. (2019) regarding the existing literature's reliance on model inductive bias, we hope the reviewer will agree that providing a proper theoretical framework for understanding weak supervision is both timely and valuable.
>
>
> On the Significance of the Synthetic Experiments
> -----------------------------------------------------------------
> The experiment section can broadly be divided into two categories:
>
> 1. Experiments to test our theoretical guarantees (Sections 6.2.1 and 6.2.3)
>
> Sections 6.2.1 and 6.2.3 primarily serve as demonstrations that the guarantees in Theorem 1 can be achieved in practice with machine learning models on synthetic datasets. Since ground truth factors are available on synthetic datasets, we used this as an opportunity to take extensive measurements to validate the various facets of our theoretical statements (Figures 3-5 and 7-16). Furthermore, the synthetic datasets that we chose are common benchmarks used in the existing disentanglement literature (Locatello et al. 2019; Burgess et al. 2018; Kim et al. 2018; Ridgeway et al. 2018; Denton et al. 2017; Mathieu et al. 2016; Watters et al. 2019). The performances of our models on existing disentanglement metrics (Figure 13) can also be directly compared with the scores reported in Locatello et al. (2019).
>
> We agree that the use of non-synthetic data would further enhance the validation of these theoretical guarantees, but we hope that this does not detract from the legitimacy and extensive nature of our existing experiments.
>
> 2. Experiments that assess the gap between theory and practice (6.2.2)
>
> Figure 4 of Section 6.2.2 addresses the following question: if Theorem 2 states that disentanglement is not guaranteed in, for example, style-content disentanglement, why have we as a community not established this as common knowledge yet?
>
> Figure 4 shows that the correlation between consistency and restrictiveness is quite strong across models tested across a broad range of hyperparameters. This finding leads us to make the following two conclusions.
>
> 1. It is easy to mistake the practical reliability of style-content disentanglement as a theoretical guarantee.
>
> 2. The practical reliability of style-content disentanglement is poorly understood and we wish to call attention to the need to provide a proper theoretical characterization of this reliability.
>
> We believe both conclusions are important in contextualizing the current literature on weakly supervised disentanglement, and hope that more researchers will be aware of these findings.
>
>
> On the Practicality of Weak Supervision Methods with Guarantees
> ----------------------------------------------------------------------------------------
> Since our paper shows examples of weak supervision methods that guarantee disentanglement, the reviewer is rightfully concerned about whether these methods can scale in practice.
>
> We believe that the scalability of these methods ultimately need to be addressed on a case-by-case basis in practice. While not within the scope of our paper, we acknowledge that the following questions are both important and challenging:
>
> 1. What is the actual cost of weak supervision methods with guarantees for a particular setting?
>
> 2. Are there settings that admit cheaper types of weak supervision methods with guarantees, compared to the specific methods analyzed in this paper?
>
> 3. In what settings are weak supervision methods (that do not have guarantees) practically reliable? And when they are reliable, why are they reliable?
>
> We believe our paper naturally inspires these important questions. If the reviewer believes it is appropriate, we would like to include these questions as concluding thoughts in the paper to facilitate further discussion on weakly supervised disentanglement.

---

> > ### Author Response · Authors · 2019-11-15
> > **Response (cont'd)**
> >
> >
> > References
> > ---------------
> >
> > Locatello et al. (2019) Challenging Common Assumptions in the Unsupervised Learning of Disentangled Representations
> > Burgess et al. (2018) Understanding disentangling in β-VAE
> > Kim et al. (2018) Disentangling by Factorising
> > Ridgeway et al. (2018) Learning Deep Disentangled Embeddings With the F-Statistic Loss
> > Denton et al. (2017) Unsupervised Learning of Disentangled Representations from Video
> > Mathieu et al. (2016) Disentangling factors of variation in deep representations using adversarial training
> > Watters et al. (2019) Spatial Broadcast Decoder: A Simple Architecture for Learning Disentangled Representations in VAEs

---

### Official Review · AnonReviewer1 · 2019-10-23
**Official Blind Review #1**

**Rating:** 8

**Review:**

This paper first discusses some concepts related to disentanglement. The authors propose to decompose disentanglement into two distinct concepts: consistency and restrictiveness. Then, a calculus of disentanglement is introduced to reveal the relationship between restrictiveness and consistency. The proposed concepts are applied to analyze weak supervision methods.

This paper is well structured. The presentation is easy to follow. The problem discussed is important to the machine learning community. The concepts discussed are supported by a large number of experiments.

The assumption that disentanglement can be decomposed into consistency and restrictiveness might be flawed. Let us consider a generator $g(Z)$  that always generates the same image for all $Z \sim p(Z)$. Note that $g(Z)$ gives perfect consistency and perfect restrictiveness as defined in Equation (3) and (6). However, we consider $g(Z)$ is a bad generator, and we do not think the corresponding latent representation $Z$ achieves perfect disentanglement. Note that such $Z$, in general, gives low values in the existing disentanglement metrics.

This implies that we might need to introduce a third component to disentanglement, which I call it relevance. We should additionally assume that different $z_{ \setminus I}$ leads to different generated images. It might be challenging to measure relevance quantitatively under the probabilistic framework, but I believe this is necessary.

In summary, I think the idea presented is interesting and useful. I believe this paper is promising and impactful after proper revision.  However, I do not recommend acceptance because it looks technically flawed.

Minor:
In Figure 5, the illustration is clear to me, but I am not sure how the vertical axis simultaneously represents two variables $z_2, z_3$.
In the table on page 5, $n$ represents the number of dimensions, right?



**Experience Assessment:**

I have published one or two papers in this area.

**Review Assessment: Checking Correctness Of Derivations And Theory:**

I assessed the sensibility of the derivations and theory.

**Review Assessment: Checking Correctness Of Experiments:**

I assessed the sensibility of the experiments.

**Review Assessment: Thoroughness In Paper Reading:**

I read the paper thoroughly.

---

> ### Author Response · Authors · 2019-11-09
> **Our paper addresses latent code informativeness (i.e. relevance)**
>
> Dear Reviewer,
>
> Thank you for acknowledging the significance of the problem being tackled in this paper. We would like to address your concern regarding the issue of degenerate generators that always output the same image for all choices of $Z$. In our paper, we referred to such a latent space $Z$ as being “uninformative”.
>
> This scenario is explicitly prevented by our requirement, stated on page 3, that our analysis is subject to the condition that “$g(Z)$ =d= $g^*(S)$ are equal in distribution”. We would like to emphasize that the decomposition of disentanglement into restrictiveness and consistency is appropriate when this condition ($g(Z)$ =d= $g^*(S)$) is satisfied, otherwise we would indeed be admitting degenerate models that have uninformative latent spaces. We addressed this point once again on page 6, stating that
>
> “The distribution matching requirement $g(Z)$ =d= $g^*(S)$ ensures latent code informativeness, i.e., preventing trivial solutions where the latent code is uninformative (see Theorem 7 for formal statement)”
>
> In the appendix, Theorem 7 formalizes what it means for the latent space to be informative and states that informativeness of the latent space is guaranteed when $g(Z)$ =d= $g^*(S)$. The formal statement and proof are available on Page 33.
>
> In other words, our proposed concepts of consistency and restrictiveness are best thought of as complementing latent code informativeness, and not as replacements for checking the informativeness of the latent code. We also wish to note that our theoretical guarantees in Theorem 1 explicitly require $g(Z)$ =d= $g^*(S)$. This is because the distribution matching setup in Theorem 1 implies distribution matching of $g(Z)$ with $g^*(S)$.
>
> As such, with respect to our theoretical analysis, we hope that you will agree that our submission takes careful measures to explicitly combat the type of degeneracy that you pointed out.
>
> In practice, when distribution matching is not guaranteed, we fully support that practitioners should check for both latent code informativeness in addition to restrictiveness and consistency. Ultimately, we note that generator restrictiveness and consistency are concepts that complement latent code informativeness, and not replacements for it. We will strive to make this point even clearer when we update the paper.
>
> ---
>
> Regarding minor comments:
> In Figure 2, the way to interpret the vertical axis is that each row corresponds to a randomly sampled choice of $(z_2, z_3)$. This is stated in the caption as, “each row denotes a fixed choice of $(z_2, z_3)$”.
>
> In the calculus, you are correct that $n$ denotes the number of dimensions. We shall make this point clear when updating our paper.

---

> > ### Comment · AnonReviewer1 · 2019-11-15
> > **Thank you for your response.**
> >
> > Thank you for your response. Now I understand that the 3 distribution matching cases in Theorem 1 imply $G(Z) \overset{d}{=} g^*(S)$, which is a prerequisite for consistency and restrictiveness.
> >
> > I suggest the authors explicitly state that the consistency and restrictiveness defined in Equations (6) and (10) are more for theoretical analysis, rather than a practical metrics for disentanglement performance; because these metrics do not measure the informativeness. Note that $G(Z) \overset{d}{=} g^*(S)$ is not likely to be true in practice. Therefore, if one uses consistency and restrictiveness metrics for model selection, it is very likely that a less informative model will be selected. In fact, I suspect the high correlation in Figure 4 $c(i)$ and $r(i)$ are due to the informativeness of the model, rather than the true relationships between consistency and restrictiveness. Because it has been observed that some disentanglement models, such as $\beta$-VAE, give worse reconstructions when better disentanglement is achieved. This might suggest that in practice, a model that achieves a better disentanglement is likely to be less informative.
> >
> > I have updated my review rating because my major concerns are addressed.

---

> > > ### Author Response · Authors · 2019-11-15
> > > **Response**
> > >
> > > Dear Reviewer,
> > >
> > > Thank you for your kind reconsideration of our paper. We will adjust our paper accordingly to emphasize that consistency and restrictiveness are primarily intended as tools for theoretical analysis.
> > >
> > > Regarding your hypothesis that using consistency and restrictiveness metrics for model selection will select for less informative models, our finding on synthetic datasets is that this is not the case (Appendix G provides visualizations of the models selected by consistency. The latent interpolations indicate that the representations are informative). As such, we believe that uninformativeness is unlikely to be a confounder for Figure 4. To verify this, we will investigate this further and check, for example, that the (high-consistency + high-restrictiveness) models in Figure 4 are not attributable to uninformativeness. We apologize for not conducting this analysis before the rebuttal deadline, but we are confident about our hypothesis and will commit to releasing this analysis when it is ready.
> > >
> > > For non-synthetic datasets where $g(Z)$ and $g^*(S)$ are likely to deviate considerably from each other, we agree that using solely consistency/restrictiveness for model selection may be biased toward uninformativeness models. One potential way to defend against uninformative models would be to estimate the mutual information between $X$ and $Z$ (which can be done in an unsupervised manner) and restrict model selection to models above a certain information threshold. Ultimately, however, we believe that model selection is best done with a proper disentanglement scoring rule, and that the development of such a scoring rule is still an open and important question.

---

### Official Review · AnonReviewer2 · 2019-10-25
**Official Blind Review #2**

**Rating:** 8

**Review:**


The paper tries to bring some theoretical foundation to the weakly supervised disentanglement. Overall it is a good contribution, but the message of the paper is not clear.  The authors propose two notions: consistency and restrictiveness, which they don't imply each other. However, the experiment on real data shows that they are highly correlated.   Up until the experiment section, the paper is well written (although a bit verbose). It seems that it is great but unfinished work.

The paper is well written, but in my opinion, there is too much verbosity on page 4-5 on rather trivial definitions consistency and restrictiveness and a big box in the calculus of disentanglement that steals space from the main results.  In my opinion, those sections can be reduced so that other theorem can be covered. In my opinion, the theorem nine should be part of the main text.

I understand the definition of "Sufficiency for Disentanglement " but it is not clear why it is important. Sure, it is a strong definition that says for any  $\mathcal{H}$ (and not a subset) the algorithm ($\mathcal{A}$) should be able to match the distribution of the observation but why is it a big deal according to the next paragraph?

I don't see any proof that Eq.11 should be between [0,1]. Yes, g is optimal, and if you enter suboptimal values to it, one expects the nominator to be less than dominator. However, g a function that is optimal in expectation, which does not mean for every s value it nominator is less than the denominator. In fact, some of the values in fig 3 are small negatives.

Fig 3 is not explained well: you are showing normalized consistency and restiveness. First of all, what is the dataset you tried this on? Second, why some values are negative?! These are supposed to be between [0,1]. Third, what is the take-home-message of this figure? the first two matrices from left show that the factors are consistent b/c they are almost diagonal. The third one from left shows that the algorithm you used is not restrictive? Then are you suggesting this as a metric of evaluation? I am not sure I understand the first figure from the right.
Overall, the authors perform a significant amount of experiments, but they did a poor job in summarizing the results.
Finally, the authors claim
                  "...We believe this correlation between consistency and restrictiveness to have been a general source of confusion in the disentanglement literature, causing many to either observe or believe that restricted labeling or share pairing on $S_i$ (which only guarantees consistency) is sufficient for disentangling Si ..."

Each of those methods should be analyzed separately to ensure that their algorithms do not induce restiveness. I just don't see the natural connection between your figure 4 and this conclusion that you made.

Minor:
Where is the proof for Theorem 1? In the Supp, it starts with Theorem 8, I guess you meant Lemma 8? You need to clean up the Supp so that one can find the proof easily. I suggest restructuring the Supp to less and finally proof of Thorem 1.

**Experience Assessment:**

I have published one or two papers in this area.

**Review Assessment: Checking Correctness Of Derivations And Theory:**

I did not assess the derivations or theory.

**Review Assessment: Checking Correctness Of Experiments:**

I assessed the sensibility of the experiments.

**Review Assessment: Thoroughness In Paper Reading:**

I read the paper thoroughly.

---

> ### Author Response · Authors · 2019-11-14
> **Response**
>
> Dear Reviewer,
>
> Thank you for your feedback! We wish to address your concerns as follows.
>
> Regarding the message of the paper
> -------------------------------------------------
> The goal of this paper is to clarify some confusion regarding the concept of disentanglement in the current literature, and to build up a theoretical framework that clarifies the guarantees actually conferred by several popular weak supervision methods
>
>
> Regarding the seeming verbosity of the definitions
> ------------------------------------------------------------------
> As evident from Figure 2, consistency and restrictiveness are distinct concepts that play critical roles in theory. We agree that there is a noticeable contrast in the simplicity of the intuition of consistency/restrictiveness versus the lengthy mathematical description. Although those concepts are intuitive, the mathematical formalism we give is carefully designed with several important subtleties in mind:
>
> 1. The specific sampling procedure in Eq (1, 2) and Eq (4, 5) are important for ensuring that the definition of disentanglement allows for correlated but interpretable features (similar to Suter et al 2018). We note that many existing definitions for disentanglement break down when the ground truth factors are correlated, and so we believe it is important to break away from the trend of making the definition of disentanglement implicitly reliant on the assumption of statistically independent ground truth factors.
>
> 2. Since consistency and restrictiveness are asymmetric concepts, generator-side consistency and restrictiveness and encoder-side consistency and restrictiveness are highly related, but not completely identical concepts. And this is the reason why Theorem 1 explicitly provides guarantee for both generator and encoder-side consistency.
> We aim to further simplify our exposition in future iterations of the paper.
>
>
> Regarding the Sufficiency for Disentanglement Formalism
> -----------------------------------------------------------------------------
> The primary significance of this section is in formalizing what is meant by particular supervision method being insufficient. Our definition of sufficiency aims to distinguish the guarantees that arise solely from the choice of supervision versus the inductive bias of architecture or the objective function. Locatello et al.’s impossibility result is restricted to a specific kind of learning algorithm (matching p_data(x) with p_theta(x)). In contrast, our definition for sufficiency is agnostic to the choice of learning algorithm and inductive bias.
>
>
> Regarding normalized consistency/restrictiveness scores
> ----------------------------------------------------------------------------
> The proof for why the score is bounded between [0, 1] directly follows from Lemma 1 in the Appendix I.2.1. We have updated the paper to make this connection explicit. Because the scores in the figure are estimated via monte carlo sampling, the monte carlo estimator is not guaranteed to be within the interval [0, 1]. However, we hope it is evident from the magnitude of the negative numbers (and from Figures 8, 9, 10) that the small negative values are noisy estimates of a score of zero.

---

> > ### Author Response · Authors · 2019-11-14
> > **Response (cont'd)**
> >
> >
> > Regarding Figure 3
> > --------------------------
> > As stated in the caption, Figure 3 is trained on Shapes3D. Figure 3 is a condensed version of a subset of the results shown in Figures 7, 8, 9, and 10. The key goal of Figure 3 is to verify our hypothesis that: restricted / share / rank pairing on a particular factor guarantees consistency on that factor, and that change pairing on a particular factor guarantees restrictiveness on that factor.
> >
> > As stated in the captions, this hypothesis predicts that the diagonal components of the heatmap should have the highest scores. And indeed, this is generally the case. Even for the third heatmap, the diagonal components of the heatmap achieve the highest scores by a statistically significant margin (Figure 8 column 1 provides additional quantile information not displayed in the heatmap).
> >
> > We believe the reviewer’s concern is moreso regarding why the margin of improvement (of diagonal v. non-diagonal scores) appears to be smaller for restrictiveness than for consistency. It is worth noting that single-factor restrictiveness measures the expected L2-discrepancy over a single factor, whereas single-factor consistency measures the expected L2-discrepancy over (n - 1) factors. This, we believe, is the main contributor to why restrictiveness scores tend to be higher than consistency scores on the off-diagonal elements. While we do not have a complete explanation for why Factors 3/4/5 for Shapes3D tended to have even higher off-diagonal restrictiveness scores than Factors 0/1/2, we note that Factors 0/1/2 control the color attributes (wall color, floor color, shape color) whereas Factors 3/4/5 control non-color attributes (object size, object shape, and azimuth).
> >
> > Regarding the right-most figure: each column represents a specific intersection experiment. For example, column 0 is trained on a weak supervision of S_{0, 1, 2} and S_{0, 3, 4, 5}. Thus, by the calculus of intersection, only consistency of factor 0 (i.e. S_0) is guaranteed.
> >
> >
> > Regarding the Use of Consistency and Restrictiveness as Metrics
> > ---------------------------------------------------------------------------------------
> > We wish to emphasize that consistency and restrictiveness are intended primarily as concepts that establish the theoretical guarantees possible with weak supervision, and not as metrics that should be used in practice for evaluating disentanglement. In our paper, we converted these concepts into metrics (i.e. normalized consistency and restrictiveness scores) solely as a means of probing the behavior the models and for checking whether the behavior is consistent with our theoretical guarantees.
> >
> > As probes, normalized consistency and restrictiveness also allow us to identify models that are consistent but not restrictiveness for a particular factor (and vice versa). As evident from Figure 4, such models do exist. We have added appendix B to illustrate two prototypical examples of models from disentanglement_lib that are consistent but not restrictive (and vice versa) that we found using our normalized consistency and restrictiveness probes across the population of 12800 pretrained models.
> >
> > While we do have some experiments demonstrating that summary statistics using consistency/restrictiveness can serve as reasonable surrogate metrics for the selection of disentangled models (see Figure 16 and Appendix F), we believe the development of a proper disentanglement scoring rule remains an open problem. We note that the primary goal of this paper is not to develop such a metric. Our goal is to clarify some confusion regarding the concept of disentanglement in the current literature, and to build up a theoretical framework that clarifies the guarantees (or lack thereof) that are actually conferred by various weak supervision methods.

---

> > > ### Author Response · Authors · 2019-11-14
> > > **Response (final)**
> > >
> > >
> > > Regarding Figure 4
> > > --------------------------
> > > Figure 4 indicates two phenomena:
> > >
> > > 1. When trained with single-factor weak supervision, examples of models that are “consistent but not restrictive” and “restrictive but not consistent” do occur. This is consistent with our theory, since restrictiveness does not imply consistency and vice versa.
> > >
> > > 2. The occurrence of models that are “consistent but not restrictive” and “restrictive but not consistent” are rare in practice.
> > >
> > > Of the weak supervision choices shown in Figure 4, we note that restricted labeling is a common approach used to achieve disentanglement in, for example, the style-content disentanglement literature. The existing literature may mislead readers into thinking that restricted labeling can guarantee disentanglement. Our theory and phenomenon (1) indicates that restricted labeling does not have this guarantee. However, phenomenon (2) indicates that examples that violate disentanglement happen infrequently in practice. Taken together, our findings are interesting because
> > >
> > > 1. We highlight that it is easy to mistake the practical reliably of style-content disentanglement as a theoretical guarantee.
> > >
> > > 2. The practical reliability of style-content is poorly understood and we wish to call attention to the need to provide a proper theoretical characterization of this reliability.
> > >
> > >
> > > Regarding Minor Feedback
> > > ------------------------------------
> > > Thank you for pointing out the theorem labeling mis-match in the appendix. We have updated the paper to fix this issue.

---

> ### Comment · AnonReviewer2 · 2019-11-15
> **updated score**
>
> I still have some concern about the practicality of the proposed method but I think it is a valuable contribution and I have updated my score accordingly.

---

> > ### Author Response · Authors · 2019-11-15
> > **Response**
> >
> > Dear Reviewer,
> >
> > Thank you for your kind reconsideration of our paper. Regarding your concerns about the practicality of the proposed methods, we wish to provide an adapted version of our response to Reviewer #3.
> >
> > ----------------------
> >
> > Regarding the practicality of the proposed methods, we wish to note that the specific weak supervision methods we analyze (such as restricted labeling) are already employed on real-world datasets, for example, in style-content disentanglement. Our contribution is thus to demonstrate the right way these methods can be used or combined to guarantee disentanglement.
> >
> > We acknowledge that the degree of supervision needed for methods to actually guarantee disentanglement might not necessarily be scalable in practice. We believe that determining when methods with disentanglement guarantees are scalable ultimately need to be addressed on a case-by-case basis in practice. Going forward, we think the following questions will be of increasing importance to the disentanglement community:
> >
> > 1. What is the actual cost of weak supervision methods with guarantees for a particular setting?
> >
> > 2. Are there settings that admit cheaper types of weak supervision methods with guarantees, compared to the specific methods analyzed in this paper?
> >
> > 3. In what settings are weak supervision methods (that do not have guarantees) practically reliable? And when they are reliable, why are they reliable?
> >
> > We believe our paper naturally inspires these important questions and hope that future researchers will study these questions through the lens of the perspective and tools developed in our paper.

---

### Public Comment · ~Weili_Nie1 · 2019-10-15
**One question about the experiments**

Thanks for the interesting work, but I have a question about the experiments in Section 6.2.3: Why is the fully-supervised GAN worse than weakly-supervised GAN, in particular on SmallNORB and Cars3D? Because, if I understand correctly, we normally we expect the fully-supervised method performs better. Also, May I ask if the authors could provide the code for the implementation of weakly-supervised GANs?

---

> ### Author Response · Authors · 2019-10-17
> **Response and link to code**
>
> Thank you for the astute inquiry about the experimental results. Indeed, we expected Full-Label GAN to outperform the weakly-supervised GANs and were also surprised to find in Figure 12 that Change and Share pairing noticeably outperformed Full-Label on a number of metrics on the Cars3D dataset. While this could be attributable to a number of possible confounders (optimization issue, architectural differences, etc), our main hypothesis is that this is because one of the factors of variation in Cars3D is car-type (with labels {0, 1, …, 182}), for which an ordinal representation in a single dimensional continuous latent variable may be inappropriate.
>
> We had chosen for simplicity to use only continuous latent variables in all of our experiments (this experimental design choice mirrors the Locatello 2019 study. We shall make this point explicit when we update the paper appendix). This means that Full-Labeling (and Rank Pairing), which imposes a specific ordering for how the car types are embedded along the latent dimension z3 (if one-indexing), may have trouble with using z3 to encode the car-type in the order prescribed by the actual labels. In contrast, Change and Share pairing do not have to adhere to the ordering prescribed by the car-type label. We believe this additional flexibility allowed for the Change/Share models to choose a more favorable ordering when casting car-type into a single-dimensional continuous representation, thus enabling better encoding of car-type into the latent variable z3.
>
> This hypothesis is consistent with our interpretation of Figure 13, which shows the normalized consistency score for each label. Notice in (Figure 13, Cars column, Row 3) that Change/Share models significantly outperformed Rank/Full-Label on normalized consistency of the car-type factor, indicating that the Rank/Full-Label models had difficulty learning a representation of car-type that is consistent with the car-type labels.
>
> As for SmallNORB, Figure 12 shows that Full-Label GAN is the top performer in 2 / 6 metrics, and a top-2 performer 5 / 6 metrics. As such, we do not think we can definitively claim that the weakly supervised GANs outperform Full-Label GAN on SmallNORB.
>
> =====
>
> We have also uploaded an anonymized version of our code to: https://drive.google.com/open?id=1VjMNOD3uFrCx4nZSgLlKrVLVbluPnGPa
>
> Please let us know if you have any trouble running it.

---

### Decision · Program_Chairs · 2019-12-19

**Decision:**

Accept (Poster)

**Comment:**

This paper first discusses some concepts related to disentanglement. The authors propose to decompose disentanglement into two distinct concepts: consistency and restrictiveness. Then, a calculus of disentanglement is introduced to reveal the relationship between restrictiveness and consistency. The proposed concepts are applied to analyze weak supervision methods.

The reviewers ultimately decided this paper is well-written and has content which is of general interest to the ICLR community.